# REVISITING THE ASSUMPTION OF LATENT SEPARABILITY FOR BACKDOOR DEFENSES

**Xiangyu Qi**[1]*, **Tinghao Xie**[1]*, **Yiming Li**[2], **Saeed Mahloujifar**[1], **Prateek Mittal**[1]
[1]Princeton University
[2]Tsinghua Shenzhen International Graduate School, Tsinghua University
{xiangyuqi,thx,sfar,pmittal}@princeton.edu; li-ym18@mails.tsinghua.edu.cn

## ABSTRACT

Recent studies revealed that deep learning is susceptible to backdoor poisoning attacks. An adversary can embed a hidden backdoor into a model to manipulate its predictions by only modifying a few training data, *without controlling the training process*. Currently, a tangible signature has been widely observed across a diverse set of backdoor poisoning attacks — models trained on a poisoned dataset tend to learn separable latent representations for poison and clean samples. This latent separation is so pervasive that a family of backdoor defenses directly take it as a default assumption (dubbed *latent separability assumption*), based on which to identify poison samples via cluster analysis in the latent space. An intriguing question consequently follows: *is the latent separation unavoidable for backdoor poisoning attacks*? This question is central to understanding whether the assumption of latent separability provides a reliable foundation for defending against backdoor poisoning attacks. In this paper, we design *adaptive backdoor poisoning attacks* to present counter-examples against this assumption. Our methods include two key components: (1) a set of trigger-planted samples correctly labeled to their semantic classes (other than the target class) that can regularize backdoor learning; (2) asymmetric trigger planting strategies that help to boost attack success rate (ASR) as well as to diversify latent representations of poison samples. Extensive experiments on benchmark datasets verify the effectiveness of our adaptive attacks in bypassing existing latent separation based defenses. Our codes are available at `https://github.com/Unispac/Circumventing-Backdoor-Defenses`.

## 1 INTRODUCTION

Overparameterized deep neural network (DNN) models can fit complex datasets perfectly and generalize well on *i.i.d.* data distributions. However, the strong capacity of these models also render them susceptible to *backdoor poisoning attacks* (Gu et al., 2017; Chen et al., 2017; Turner et al., 2019; Li et al., 2022). In a backdoor poisoning attack, an adversary only manipulates a small portion of the victim's training dataset. The victims will train their own model on the manipulated dataset and consequently get a backdoored model. Typically, the adversary will *poison* the victim's dataset by injecting a small amount of backdoor poison samples, each of which contains a backdoor trigger (e.g. a specific pixel patch) and is labeled to a specific target class. A DNN model trained on this poisoned dataset will be backdoored in that they tend to learn an artificial correlation between the backdoor trigger and the target class. These attacks are stealthy since backdoored models behave normally on natural samples and therefore users can hardly identify them.

Despite the stealthiness in terms of model performance on natural samples, it has been commonly observed (Tran et al., 2018; Chen et al., 2019; Huang et al., 2022) that backdoor poisoning attacks tend to leave tangible signatures in the latent space of backdoored models. As visualized in Fig 1b - Fig 1g, poison and clean samples from the target class consistently form two separate clusters in the latent space, across a diverse set of backdoor poisoning attacks. The pervasiveness of the latent separation renders itself oftentimes as a default assumption, which we call *latent separability assumption* in this work. A family of defenses (*i.e.*, *latent separation based backdoor defenses*) explicitly base their designs on this assumption. These defenses first train a base classifier on the

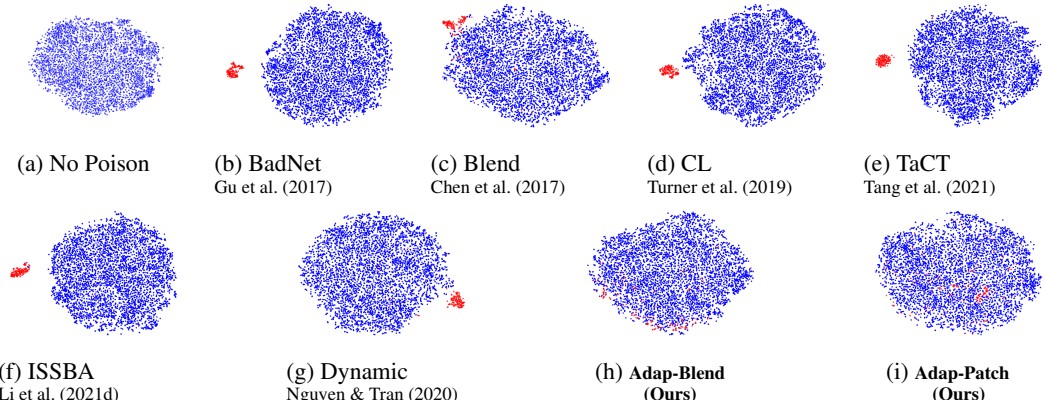

Figure 1: T-SNE visualization of latent separability characteristic on CIFAR-10. Each point in the plots corresponds to a training sample from the target class. Caption of each subplot specifies its corresponding poison strategy. To highlight the separation, all poison samples are denoted by red points, while clean samples correspond to blue points.

poisoned dataset, and expect the base model will naturally learn separable latent representations for poison and clean samples respectively. After that, they perform cluster analysis on the latent space of the base model. If the latent separation characteristics reliably arise, these defenses will be able to identify the outlier cluster formed by poison samples, and thus accurately filter out these poison samples from the training set. We note that *this family of defenses are particularly important and successful in the backdoor defense literature*. Popular proposals in this family like Spectral Signature (Tran et al., 2018) and Activation Clustering (Chen et al., 2019) have already become indispensable baselines, and recent state-of-the-art proposals including SCAn (Tang et al., 2021) and SPECTRE (Hayase et al., 2021) in this family even claim to achieve nearly perfect recall with negligible false positive rate against a diverse set of attacks. Given the pervasiveness of the latent separation and its profound success in the application of backdoor defenses, a natural question arises: *Is the latent separation unavoidable for backdoor poisoning attacks?*

In this work, we revisit the assumption of latent separability and expose failure regions of defenses based on it. Specifically, we design adaptive backdoor poisoning attacks (without control of the model training process), which can actively suppress the latent separation while maintaining a high attack success rate (ASR) with negligible clean accuracy drop. Two critical components are underlying the design of our adaptive attacks (see Fig 2 for an overview): (1) *Data poisoning based regularization*. After planting the backdoor trigger to a set of samples, we do not mislabel all of them to the target class. Instead, we randomly keep a fraction of them (namely regularization samples) still correctly labeled to their real semantic classes. Intuitively, these additional regularization samples penalize the backdoor correlation between the trigger and the target class. (2) *Trigger planting strategies that promote asymmetry and diversity*. One may notice that penalization on the backdoor correlation induced by regularization samples can also greatly hurt the attack success rate (ASR). We alleviate this problem via asymmetric trigger planting strategies. As illustrated in Fig 2, we apply weakened triggers when we construct regularization and payload samples for data poisoning, while the original standard trigger would only be used during test time to activate the backdoor. Conceptually, in this way, since test-time backdoor samples (with the standard trigger) contain stronger backdoor features than those of regularization samples (with weakened triggers), the test-time attack can well mitigate the counter force from regularization samples and still maintain a high ASR. Besides asymmetry, our design also promotes diversity of triggers during data poisoning — different poison samples could be stamped with different partial triggers, selected from a diverse set of trigger partitions. Intuitively, this diversity allows backdoor poison samples to scatter more diversely in the latent representation space, and can thus avoid being aggregated into an easy-to-identify cluster.

In conclusion, the main contributions of this paper are four-fold. **(1)** We confirm that the latent separability assumption holds across a diverse set of backdoor poisoning attacks in the existing literature. **(2)** We reveal that this assumption could fail, leading to poor performance of defenses that explicitly base their designs on it. **(3)** We design some simple yet effective adaptive backdoor poi-

---

*The first two authors contributed equally to this paper.
Correspondence to: Xiangyu Qi, Tinghao Xie, Yiming Li, and Prateek Mittal.

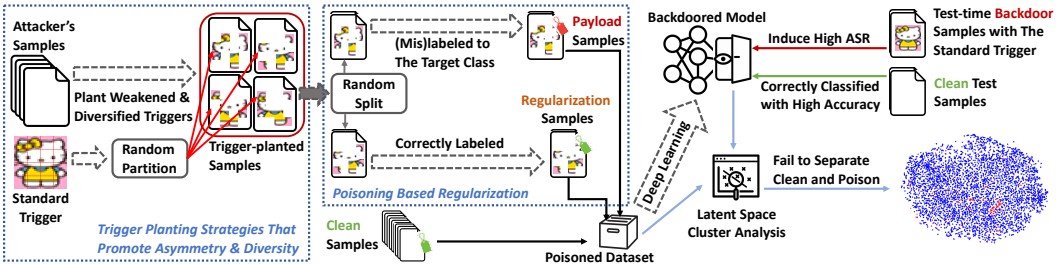

Figure 2: An overview of our adaptive backdoor poisoning attacks (here we take Adaptive-Blend introduced in Section 5.2 as an example for illustration). Two key components render our attacks adaptive: **(1)** Poisoning-based regularization, which penalizes the backdoor correlation and helps to suppress the latent separation; **(2)** Trigger planting strategies that promote asymmetry and diversity, which help to maintain a high attack success rate as well as to improve latent space stealthiness. Please refer to our Section 5.1 for more technical details.

soning attacks to present counter-examples against this assumption with two key novel components. **(4)** We conduct extensive experiments on benchmark datasets, verifying the effectiveness and the stealthiness in countering detection methods of our adaptive attacks.

## 2  RELATED WORK

**Backdoor Poisoning Attacks.**   Backdoor poisoning attacks (Gu et al., 2017; Chen et al., 2017; Turner et al., 2019; Li et al., 2021d) are also frequently referred to as poison-only backdoor attacks. This category of attacks only assume control over a small portion of the victim's training data, while the victim will train her own models on the poisoned dataset from scratch. Other backdoor attacks that assume additional control over the training process (Shokri et al., 2020) or even weights of deployed models (Liu et al., 2017a; Qi et al., 2021; 2022a) do not fall in this category and are not considered in this work. We refer readers to Li et al. (2022) for a more comprehensive review.

**Latent Separation for Backdoor Defenses.**   It has been commonly observed (Tran et al., 2018) that models trained on a poisoned dataset tend to learn very different latent representations for backdoor and clean samples in the target class, which form two separate clusters (see Fig 1). This phenomenons is so pervasive that a family of defenses directly take the latent separation as a default assumption and propose to identify poison samples via performing cluster analysis on the latent space. This family includes Spectral Signature (Tran et al., 2018) and Activation Clustering (Chen et al., 2019), which are most commonly evaluated baselines. More recent proposals (Tang et al., 2021; Hayase et al., 2021) in this family further claim to achieve nearly perfect recall with negligible false positive rate against a diverse set of attacks even in very low poison rate cases.

**Adaptive Backdoor Attacks Against Latent Separation Based Defenses.**   A family of adaptive backdoor attacks (Tan & Shokri, 2020; Xia et al., 2022; Doan et al., 2021; Ren et al., 2021; Cheng et al., 2021; Zhong et al., 2022) explicitly aim to reduce the latent separation between poison and clean samples. However, they do not fit into the paradigm of backdoor poisoning attacks — they assume additional control over the whole training process and thus directly encode the latent inseparability into the training objectives of attacked models. A more relevant work is Tang et al. (2021), which points out that their source-specific poison-only attack can reduce latent separation. However, as shown in Fig 1e, when the base model is trained along with standard data augmentation, *there is still a notable separation* between the clean and poison populations, and actually Tang et al. (2021) themselves also show that an improved latent space cluster analysis suffices to perfectly separate poison and clean samples of this attack. *Thus, it is still unclear whether a poison-only backdoor attack can overcome the latent separation to evade backdoor defenses built on it.* In this work, we fill the gap and design adaptive backdoor poisoning attacks that can actively suppress the latent separation (and thus circumvent existing latent separation based defenses).

**Other Backdoor Defenses.**   There are other defenses that are not built on latent separation. These include trigger synthesis (Wang et al., 2019; Guo et al., 2022), model diagnosis (Xu et al., 2021; Kolouri et al., 2020), sample diagnosis (Gao et al., 2019; Guo et al., 2023), fine-tuning (Liu et al., 2017b; Li et al., 2021b), poison suppression (Li et al., 2021a; Huang et al., 2022) and proactive training (Qi et al., 2022b), etc. Many of these proposals also have their own limitations revealed by existing literature (refer Li et al. (2022)), but they are not our focus in this work.

## 3 NOTATIONS AND THREAT MODEL

**Notations.** We study image classification with DNN models. We denote a model by $\mathcal{F}_\theta : \mathcal{X} \mapsto [C]$, where $\theta$ are trainable parameters, $\mathcal{X}$ is the input space, $C$ is the number of classes, and $[C] := \{1, 2, \ldots, C\}$. We decompose $\mathcal{F}_\theta$ as $\mathcal{F}_\theta = l_\theta \circ f_\theta$, where $l_\theta$ is the last linear prediction layer that transforms a latent representation into the final prediction label, and $f_\theta$ is the feature extractor. Given an input $x \in \mathcal{X}$, $f_\theta(x) \in \mathcal{H}$ is the latent representation of $x$ $w.r.t$ model $\mathcal{F}_\theta$, $\mathcal{H}$ denotes the latent representation space, and $\mathcal{F}_\theta(x) = l_\theta \circ f_\theta(x)$ is the predicted label. For backdoor poisoning attacks, we denote the clean training set by $\mathcal{D} = \{(x_i, y_i) \mid i = 1, \ldots, n\}$. We denote the backdoor trigger planting strategy by a transformation $\mathcal{T} : \mathcal{X} \mapsto \mathcal{X}$, and the adversary's poison label flipping strategy is denoted by $\mathcal{L} : \mathcal{X} \times [C] \mapsto [C]$. We use $\mathcal{J} := \{j_1, \ldots, j_p\}$ to denote indices of the $p$ data points that are controlled by the adversary. The resulting poisoned training set is denoted as $\mathcal{D}_{\text{poison}} = \{(\tilde{x}_i, \tilde{y}_i) \mid i = 1, \ldots, n\}$, where

$$\tilde{x}_i = \begin{cases} \mathcal{T}(x_i), & i \in \mathcal{J} \\ x_i, & \text{otherwise} \end{cases}, \qquad \tilde{y}_i = \begin{cases} \mathcal{L}(x_i, y_i), & i \in \mathcal{J} \\ y_i, & \text{otherwise} \end{cases}. \qquad (1)$$

**Threat Model.** We consider the standard threat model of backdoor poisoning attacks (poison-only backdoor attacks), where the adversary *only* control a small portion of the victim's training data and the victim will *train her own models from scratch* on the poisoned dataset manipulated by the adversary. Specifically, the adversary will design a trigger planting strategy $\mathcal{T}$ and a label flipping strategy $\mathcal{L}$ to manipulate the controlled $p$ training samples (as formulated in Eqn 1). A victim model trained on the poisoned dataset $\mathcal{D}_{\text{poison}}$ will be backdoored — that is, during test time, the model will (mis)classify a trigger-planted input to a target class $t$ with high probability, while keeping approximately the same performance to that of a benign model on genuine inputs.

## 4 PROBLEM FORMULATION: TOWARDS POISON-ONLY BACKDOOR ATTACKS THAT CAN ACTIVELY SUPPRESS THE LATENT SEPARATION

**Latent Separability Assumption for Backdoor Defense.** Given a poisoned dataset $\mathcal{D}_{\text{poison}}$, one can train a backdoored model $\mathcal{F}_\theta := l_\theta \circ f_\theta$ via running a standard empirical risk minimization procedure $h$ on $\mathcal{D}_{\text{poison}}$, $i.e.$, $\theta \in h(\mathcal{D}_{\text{poison}})$. Latent separability assumption indicates that, in the latent representation space generated by the backdoored model $\mathcal{F}_\theta$, poison and clean samples from the target class $t$ will form separate clusters, while samples from a non-target class only form a single homogeneous cluster (see Fig 1). Latent separation based backdoor defenses (Tran et al., 2018; Chen et al., 2019; Hayase et al., 2021; Tang et al., 2021) propose to run cluster analysis on $H^c = \{f_\theta(\tilde{x}_i) \mid \tilde{y}_i = c\}$ for each class c. Typically, the defender will design a heterogeneous criterion $\mathcal{I}(\cdot)$ that takes $H^c$ as input and judges whether this set is heterogeneous ($i.e.$, contains separate clusters). On the heterogeneous $H^t$ identified by the criterion $\mathcal{I}$, the cluster analysis will divide $H^t$ into two empirical clusters $H^t_B$ and $H^t_A$, where $H^t_A$ is the suspected cluster formed by poison samples. The dataset will be cleansed by simply removing those training samples that generate $H^t_A$.

**Our Goals.** This work revisits the assumption of latent separability for backdoor defenses against poison-only backdoor attacks. We investigate *adaptive backdoor poisoning attacks* that can actively suppress the latent separation between poison and clean samples. Ideally, against such adaptive attacks, the criterion $\mathcal{I}$ used by a defense should fail to detect the heterogeneity in $H^t$ and the cluster analysis would neither accurately separate poison and clean samples.

**Perspectives that Motivate Our Design.** Two heuristic and mutually complementary perspectives on the latent separation phenomenon have inspired our design in this work. The first perspective attributes the latent separation to *the dominant impact of backdoor triggers* (Tran et al., 2018) during the inference of backdoored models. The intuition is — in order to "push" a (trigger-planted) backdoor poison sample from its semantic class to the target class, a backdoored model tends to learn an excessively strong signal for the backdoor trigger pattern in latent representation space such that the signal can overwhelmingly beat other semantic features to make its dictatorial decision. The strong backdoor signal that exclusively appears in backdoor poison samples thus leads to the separation. The second perspective is that, backdoored models learn separate representations for poison and clean samples simply because they tend to *learn a separate shortcut rule* (Geirhos et al., 2020) (solely based on the trigger pattern) to fit those poison samples without using any semantic features. The sense is — backdoor learning is often independent of (or only weakly correlated to) the semantic features used by the main task, thus the backdoored model that fits the poisoned dataset essentially just learns two unrelated (or weakly related) tasks. From this aspect, there is not even an

appealing reason for backdoor models to learn homogeneous latent representations for samples from the two heterogeneous tasks. Motivated by these perspectives, we conceive that a desirable adaptive backdoor poisoning attack (that can mitigate the latent separation) might need to encode some form of regularization, so as to (1) penalize the backdoored model for learning abnormally strong signals for the backdoor trigger; (2) encourage interconnection between backdoor learning and learning of the main task. These intuitions finally lead to our design in Section 5.

## 5 OUR METHODS

We design *adaptive* backdoor poisoning attacks following the insights we introduce in Section 4. In Section 5.1, we first present the generic framework underlying the design of our attacks. Then, in Section 5.2, we elaborate concrete attacks that we implement in this work.

### 5.1 A GENERIC FRAMEWORK FOR ADAPTIVE BACKDOOR POISONING ATTACKS

**Overview.** We present an overview of our design in Fig 2. Unlike typical backdoor poisoning attacks, in our framework, we do not label all trigger-planted samples to the target class. As shown, after planting the backdoor trigger to a set of samples (sampled from all classes), we randomly split them into two disjoint groups. For one group, we still label them to the target class (we call this group *payload samples*) to establish the backdoor correlation between the trigger pattern and the target label; while the other group (namely *regularization samples*) will instead be correctly labeled to their real semantic classes (that can be diffident from the target class) to regularize the backdoor correlation. Formally, following our notations in Section 3, the adversary will specify a *conservatism ratio* $\eta \in [0, 1)$, with which our label flipping strategy formulates as:

$$\mathcal{L}(x_i, y_i) = \begin{cases} t, & \text{with probability } 1 - \eta \\ y_i, & \text{with probability } \eta \end{cases}. \tag{2}$$

Moreover, we introduce ideas of asymmetry and diversity into our trigger design — we apply a diverse set of weakened triggers to construct regularization and payload samples for data poisoning, while the original standard trigger is used during test time to activate the backdoor.

**Regularization Samples.** We note that, *the introduction of regularization samples well incorporates our two insights from Section 4*. First, with regularization samples, the backdoored model can no longer learn a dominantly strong signal for the backdoor trigger that dictatorially votes for the target class, otherwise, it can not fit regularization samples that are correctly labeled to other classes. This explains the naming of *regularization samples* — intuitively, they serve as regularizers that help to penalize the backdoor signal in the learned latent representations. Second, the model can neither fit all trigger-planted samples via a simple shortcut rule. Instead, now *it has to fit a much more complicated boundary* that should decide when to classify a trigger-planted input to the target class and when to classify it to its real semantic label, where the boundary is randomly generated. To successfully fit this boundary, the model must rely on both the trigger pattern and artifacts from the semantic features that coexist with the trigger, thus the learned latent representations for backdoor samples should be a more balanced fusion of both the trigger pattern and semantic features.

**Asymmetric Triggers.** *The introduction of asymmetric triggers is critical for our attacks to still maintain a high attack success rate (ASR).* As one may easily notice, since regularization samples penalize the backdoor correlation, a side-effect could be the drop of attack success rate (ASR). To mitigate this problem, rather than using the same trigger for both data poisoning and test-time attack, in our design, we apply weakened triggers for data poisoning and use the (stronger) original standard trigger only for the test time. The intuition is: During test time, the backdoor samples (with the standard trigger) contain stronger backdoor features than those of regularization samples (with weakened triggers). This then enables test-time backdoor samples to have sufficient "power" to mitigate the counter force from regularization samples and thus to still achieve a high ASR. We note that the idea of asymmetric triggers traces earliest back to Chen et al. (2017), however the context is different. In order to evade human inspection on the poisoned dataset, Chen et al. (2017) propose to use weakened triggers that are visually less evident for data poisoning, and point out that a high ASR can still be maintained if the original standard trigger is used in test time. In our context, we use weakened triggers mainly to undermine the negative impact induced by regularization samples.

**Trigger Diversification.** We also highlight that *the trigger diversification in our design can also help our attacks to mitigate the latent separation.* Intuitively, since different poison samples could be planted with different triggers, these poison samples may scatter more diversely in the latent

representation space. We expect such a more diverse scattering can prevent these poison samples from aggregating into an easy-to-identify cluster.

## 5.2 INSTANTIATIONS OF OUR ATTACKS

Note that, our framework presented in Fig 2 is generic and can be creatively combined with existing techniques to instantiate powerful adaptive attacks. Following this framework, we instantiate two concrete attacks via directly adapting commonly used image blending based and patch based poison strategies, namely *Adaptive-Blend* and *Adaptive-Patch* respectively.

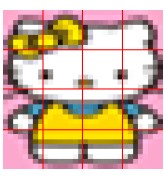 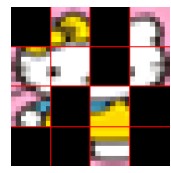

Figure 3: In Adaptive-Blend, we partition the full trigger image into $4 \times 4 = 16$ pieces (left), and randomly apply only $50\%$ of these trigger pieces (right) to each poison sample, during data poisoning. Red lines demonstrate the grids by which we randomly mask the original trigger.

**Adaptive-Blend.** An interesting point revealed by Fig 1 is that the simple Blend attack (Chen et al., 2017) turns out to induce the least latent separation, better than many attacks that are usually deemed more advanced and stealthy. This suggests image blending based triggers as good candidates for designing attacks with weak latent separation. For this reason, we design *Adaptive-Blend* via directly adapting the naive Blend attack according to our framework. Specifically, Adaptive-Blend introduces a conservatism ratio of $\eta = 0.5$ to balance the number of payload and regularization samples, and still adopts the process $\tilde{x} = (1 - \alpha) \cdot x + \alpha \cdot T$ from Chen et al. (2017) to blend the trigger pattern $T$ into a genuine image $x$ to construct the trigger-planted sample $\tilde{x}$. As for asymmetric design, we still take the standard $\alpha = 0.2$ for test-time attacking, but use a weaker asymmetric opacity of $\alpha = 0.15$ for poison samples. Moreover, for a stronger and more diverse asymmetry, we propose to partition the standard full trigger (Fig 3, left) into $4 \times 4 = 16$ pieces — the full trigger would still be used for test-time attacking, while we randomly apply only $50\%$ of the partitioned trigger pieces (*e.g.*, Fig 3, right) to each poison sample during data poisoning. This additional partition boosts both the ASR and the latent space stealthiness (see Section 6.3.3).

**Adaptive-Patch.** Although the triggers could be more visually detectable, for comprehensiveness, we also instantiate our adaptive attack with patch based triggers, namely Adaptive-Patch. Empirically, since patch based triggers usually induce stronger latent separation (*e.g.*, see Fig 1b), we correspondingly turn to a larger conservatism ratio of $\eta = 2/3$. For trigger planting, rather than sticking to a single patch pattern, we prepare a more diverse set of 4 patch triggers (Fig 6c-6f) for data poisoning. Specifically, each poison sample is randomly attached to only one of the four triggers with a low opacity (*e.g.*, $50\%$) (Fig 6j-6m). At test time, we asymmetrically apply two (of the four) fully opaque triggers (*e.g.*, Fig 6g) simultaneously to achieve high ASR.

## 6 EXPERIMENTS

### 6.1 MAIN SETTINGS

**Datasets and Model Architectures.** We evaluate our adaptive attacks on three benchmark datasets that are commonly used in backdoor learning literature: CIFAR-10 (Krizhevsky, 2012), GT-SRB (Stallkamp et al., 2012) and a 10-classes subset of Imagenet (Russakovsky et al., 2015). For building base models, we also consider three different architectures including ResNet-20 (He et al., 2016), VGG-16 (Simonyan & Zisserman, 2014) and Mobilenet-V2 (Sandler et al., 2018). Due to the space limit, in this section, we only present our results on CIFAR-10 with ResNet-20. We refer interested readers to Appendix B for results on other datasets and architectures. Detailed configurations on dataset split and training details of base models are deferred to Appendix A.

**Attacks.** We evaluate our `Adap-Blend` and `Adap-Patch` attacks presented in Section 5.2. We compare our adaptive attacks with six representative attacks in the literature. These attacks correspond to a diverse set of poisoning strategies including both classical and advanced ones. `BadNet` (Gu et al., 2017) and `Blend` (Chen et al., 2017) correspond to typical dirty-label attacks with patch-like triggers and blending based triggers respectively. `Dynamic` (Nguyen & Tran, 2020) and `ISSBA` (Li et al., 2021d) correspond to input-aware backdoor attacks. `CL` (Turner et al., 2019) is a clean label attack. `TaCT` (Tang et al., 2021) is a source-specific attack. Unless explicitly specified, for every attack, by default, we use 150 (payload) poison samples for data poisoning. Detailed attack configurations are described in Appendix A.2.

Table 1: Latent separability based defenses against our adaptive attacks on CIFAR-10.

| | (%) | No Poison | Blend | BadNet | ISSBA | Dynamic | CL | TaCT | **Adap-Blend (Ours)** | **Adap-Patch (Ours)** |
|---|---|---|---|---|---|---|---|---|---|---|
| Without Defense | ASR | / | 89.0 | 99.9 | 95.3 | 97.5 | 93.6 | 96.5 | 76.5 | 97.5 |
| | Clean Accuracy | 92.0 | 91.7 | 91.5 | 91.6 | 91.8 | 92.1 | 91.6 | 91.5 | 91.5 |
| Spectral Signature Tran et al. (2018) | Elimination Rate | / | 53.8 | 98.0 | 63.5 | 87.8 | 94.4 | 62.9 | 13.3 | 10.0 |
| | Sacrifice Rate | 15.0 | 4.4 | 4.2 | 4.3 | 4.3 | 4.2 | 4.3 | 4.5 | 4.5 |
| | ASR | / | 58.6 | 1.3 | 1.1 | 72.4 | 40.8 | 62.0 | 62.0 | 93.1 |
| | Clean Accuracy | 90.9 | 91.5 | 91.4 | 91.5 | 86.1 | 91.7 | 91.6 | 91.5 | 91.5 |
| Activation Clustering Chen et al. (2019) | Elimination Rate | / | 0.0 | 100.0 | 0.0 | 30.6 | 33.3 | 33.1 | 0.0 | 0.0 |
| | Sacrifice Rate | 0.0 | 0.0 | 7.1 | 0.0 | 5.3 | 1.0 | 5.4 | 0.0 | 0.0 |
| | ASR | / | 87.8 | 1.1 | 95.3 | 69.7 | 62.2 | 65.2 | 76.0 | 97.5 |
| | Clean Accuracy | 92.0 | 91.7 | 91.4 | 91.6 | 91.5 | 92.1 | 91.6 | 91.6 | 91.5 |
| SCAn Tang et al. (2021) | Elimination Rate | / | 0.0 | 99.1 | 91.8 | 62.9 | 66.7 | 100.0 | 0.0 | 0.0 |
| | Sacrifice Rate | 0.0 | 0.0 | 3.5 | 0.9 | 0.0 | 4.0 | 4.9 | 1.2 | 0.0 |
| | ASR | / | 87.8 | 1.0 | 0.9 | 46.3 | 32.9 | 0.5 | 78.2 | 97.5 |
| | Clean Accuracy | 92.0 | 91.7 | 91.1 | 91.6 | 91.7 | 91.7 | 90.8 | 91.6 | 91.5 |
| SPECTRE Hayase et al. (2021) | Elimination Rate | / | 96.4 | 100.0 | 100.0 | 99.8 | 100.0 | 100.0 | 6.9 | 0.0 |
| | Sacrifice Rate | 1.5 | 0.2 | 0.2 | 0.2 | 0.2 | 0.2 | 0.2 | 0.5 | 0.5 |
| | ASR | / | 5.7 | 0.8 | 1.0 | 7.7 | 1.6 | 1.7 | 69.0 | 94.8 |
| | Clean Accuracy | 91.6 | 91.7 | 91.7 | 91.6 | 91.6 | 91.6 | 91.6 | 91.4 | 91.6 |
| Silhouette Score | | / | 0.2608 | 0.4744 | 0.3933 | 0.4358 | 0.3964 | 0.2866 | **0.1065** | **0.0856** |

**Defenses.** To validate the "adaptiveness" of our attacks against latent separation based backdoor defenses, we evaluate the four state-of-the-art defenses from this family: Spectral Signature (Tran et al., 2018), Activation Clustering (Chen et al., 2019), SCAn (Tang et al., 2021) and SPEC-TRE (Hayase et al., 2021). All of these defenses are designed to detect and eliminate backdoor poison samples from the poisoned dataset, based on the assumed latent separation characteristics.

**Metrics.** For backdoor defenses that we evaluate, we measure their: 1) *Elimination Rate*, ratio of (payload) poison samples that they successfully detect; 2) *Sacrifice Rate*, ratio of clean samples falsely eliminated; 3) *Attack Success Rate (ASR)* of models retrained on the cleansed set; 4) *Clean Accuracy* of models retrained on the cleansed set. Note that, ASR is defined as the ratio of trigger-planted samples that are mispredicted to the target class, while clean accuracy is the accuracy on genuine test samples. Moreover, to quantify the latent separation between clean and poison samples, we report the *Silhouette Score* (Rousseeuw, 1987) of latent representations in the target class. A silhouette score is in the range from 0 to 1. A lower silhouette score indicates weaker separation. All the numbers that we report are average results across three independent repeated experiments.

## 6.2 MAIN RESULTS

**Visualization.** Fig 1 plots latent representations of poison and clean samples for different attacks, visualized by T-SNE (Van der Maaten & Hinton, 2008). As shown, notable latent separations are consistently observed on all the baseline attacks that we consider, while the poison and clean samples of our attacks mix with each other (Fig 1h,1i). To further reveal the extent of latent inseparability of our adaptive attacks, we also use Support Vector Machine (SVM (Cortes & Vapnik, 1995)) to find the (linear) boundary that best separates poison and clean samples in the latent representation space. Fig 4 visualizes the histogram of distances between each data point and the SVM hyperplane. As shown, compared to non-adaptive attacks (Fig 4a and 4c), our adaptive attacks (Fig 4b and 4d) bring the poison and clean samples much closer.

**Against Latent Separation Based Defenses.** We present our main results in Table 1. As shown, against SPECTRE (Hayase et al., 2021) defense, the strongest latent separation based defense in the literature, none of the six baseline attacks survive — SPECTRE can always eliminate almost all poison samples with negligible sacrifice of clean samples. This is consistent with our (qualitative) visualization in Fig 1, where notable latent separations are observed for all these attacks. It is also consistent with our quantitative measure of the latent separation — all the six baseline attacks induce high Silhouette scores ($> 0.25$). In comparison, our two adaptive attacks exhibit evident stealthiness in the latent representation space — both the visualization results (Fig 1h,1i) and Silhouette scores indicate much weaker latent separation, and all the four latent separation based defenses are consistently defeated (ASR is still larger than $20\%$ after defense). Besides, our adaptive attacks always achieve high ASR with negligible clean accuracy drop in all the cases. When no defense is applied, both Adap-Blend ($>75\%$) and Adap-Patch achieve high ASR ($>95\%$). While none of the other six baseline attacks makes thorough all these defenses after cleansing and retraining, both our Adap-Blend and Adap-Patch consistently retain considerable ASR (Adap-Blend $> 60\%$ and Adap-Patch $> 90\%$), surviving each of them. We point out that, our results serve as counter examples against

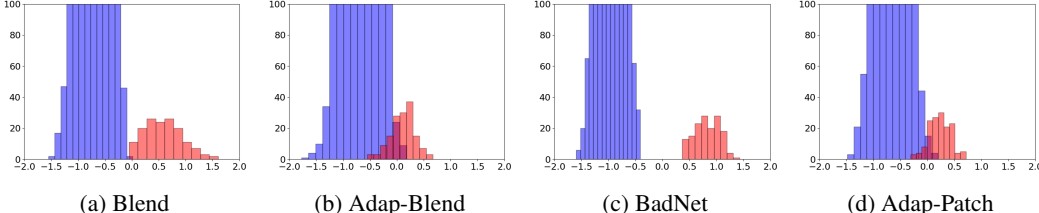

| (a) Blend | (b) Adap-Blend | (c) BadNet | (d) Adap-Patch |

Figure 4: Visualization of latent representation spaces fitted by SVM. We use SVM to find the optimal (linear) boundary that separates poison and clean samples, and plot the histograms of (signed) distances between each point and the SVM hyperplane.

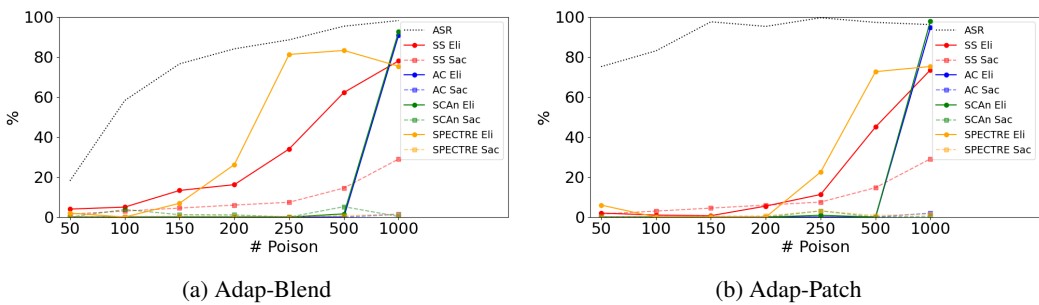

| (a) Adap-Blend | (b) Adap-Patch |

Figure 5: Defense results w.r.t. different (payload) poison samples. For Adap-Blend, we use as many regularization samples as payload samples; for Adap-Patch, we use twice the regularization sample number as payload samples. The black dotted lines show the ASR. We use different colors to represent the results of different defenses, where the solid lines correspond to Elimination ("Eli") and the dotted lines correspond to Sacrifice ("Sac").

Table 2: Adap-Blend with different regularization sample numbers, with fixed 150 payload samples.

| (%) | # Regularization Samples | 0 | 50 | 100 | 150 | 200 | 250 | 300 | 350 | 400 | 450 |
|---|---|---|---|---|---|---|---|---|---|---|---|
| | ASR | 89.0 | 86.5 | 83.9 | 76.5 | 74.1 | 70.4 | 65.6 | 60.9 | 58.0 | 56.5 |
| | Elimination Rate | 67.3 | 45.3 | 37.1 | 6.9 | 0.0 | 0.0 | 0.0 | 0.0 | 0.0 | 0.0 |
| SPECTRE | Sacrifice Rate | 0.2 | 0.3 | 0.3 | 0.5 | 0.5 | 0.5 | 0.5 | 0.5 | 0.5 | 0.5 |
| | ASR | 9.1 | 29.2 | 37.9 | 69.0 | 78.2 | 68.5 | 57.9 | 68.7 | 64.0 | 62.8 |

the assumption of latent separability and the "adaptiveness" of our attacks are also validated. We also supplement results on other datasets and model architectures (see Appendix B).

## 6.3 ABLATION STUDIES

### 6.3.1 POISON RATE

As mentioned in Section 6.1, our main experiments consistently use 150 (payload) poison samples for poisoning attack. In Fig 5, we supplement additional results of our adaptive attacks with different number of poison samples. Specifically, we increase the number of payload poison samples from 50 to 1,000, and the number of regularization samples also proportionally vary according to the fixed conservatism ratio $\eta$ that we specify in Section 5.2. As shown, one key takeaway is — when the poison sample number grows too large (*e.g.*, 1,000), the stealthiness of our adaptive attacks start to significantly degrade. This is not surprising — with more and more poison samples containing the rigid trigger pattern, the trigger pattern would become increasingly statistical significant, and models will unavoidably learn strong signal for this pattern in spite of the regularization. This indicates that a moderate poison rate is also a necessary condition for the success of our adaptive attacks.

### 6.3.2 STRENGTH OF REGULARIZATION

Now, we fix the number of payload poison samples (150 samples), and investigate varying number of regularization samples — this reflects different strength of regularization. Specifically, we evaluate Adap-Blend against SPECTRE, and present the results in Tab 2. We can generally tell that: 1) when

Table 3: Ablation study to see if every part of our adaptive strategy is necessary.

(a) Blending attack with lower poison rates.

| (%) | ASR | SPECTRE | |
|---|---|---|---|
| # Poison Samples | | Elimination | Retrained ASR |
| 50 | 69.7 | 84.0 | 8.7 |
| 100 | 81.4 | 97.0 | 5.3 |
| 150 | 89.0 | 96.4 | 5.7 |

(b) Adap-Blend with partial components.

| (%) | ASR | SPECTRE | |
|---|---|---|---|
| | | Elimination | Retrained ASR |
| No Diversity & Asymmetry | 52.1 | 28.0 | 32.5 |
| No Regularization Samples | 89.0 | 67.3 | 9.1 |
| With Both | 76.5 | 6.9 | 69.0 |

the regularization is weak (*e.g.*, 0, 50, 100 regularization samples), our adaptive attacks could still be detected; 2) when the regularization is becoming stronger, our adaptive attacks start to mitigate the defense, though the ASR suffers from more sacrifice.

### 6.3.3 Is every part of our adaptive strategy necessary?

**Simply reducing poison rate is not enough.** In Section 6.3.1, we reveal that a low poison rate is necessary for the success of our attacks. Nonetheless, as shown in Tab 3a, when we lower the poison rate (to as few as 50 poison samples) of blending attack, it will still be cleansed by SPECTRE. Thus, simply reducing poison rate is not sufficient for mitigating the latent separation.

**Simply relying on regularization samples is not enough.** Regularization samples are important in our design, so is the trigger planting strategy we adopted (See Section 5.1 for discussion). If we use the standard symmetric trigger for both the data poisoning and test-time attack, both the ASR and latent space stealthiness would degrade — the first row of Tab 3b shows Adap-Blend without asymmetric trigger partitioning, where it has a lower ASR ($\approx 50\%$) and could be further suppressed (retrained ASR $\approx 30\%$) by SPECTRE.

**Simply relying on asymmetric and diversified triggers is not enough.** Reversely, we study how our adaptive strategy behaves when we don't use regularization sample and rely solely on our trigger planting strategy. As shown in the second row of Tab 3b, though the original ASR gets higher, as a trade off, a much larger fraction ($> 67\%$) of the poison samples can now be recognized and removed by SPECTRE and the retrained ASR drops severely. This further confirms that regularization samples are vital for our attacks. See Fig 9 for a further $w.r.t.$ the necessity of regularization.

## 7 Discussions

We expect an ideal adaptive attack can make the poison and clean samples completely indistinguishable in the latent space. This has been achieved under stronger threat model where the the training process is controlled (Shokri et al., 2020; Xia et al., 2022; Doan et al., 2021; Ren et al., 2021; Cheng et al., 2021; Zhong et al., 2022). In this paper, we take a step to this goal under poisoning-only threat model. We propose adaptive backdoor poisoning attacks that can suppress the latent separation and circumvent existing defenses based on it. However, as shown in Fig 4, under oracle visualization, there is still a difference between poison and clean distributions, though the difference is greatly reduced. A remaining question is — is it possible to achieve the ideal indistinguishable goals under the poison-only setting? We encourage future work to look into this question.

## 8 Conclusion

In this work, we revisited the assumption of latent separability for backdoor defenses. We revealed that this assumption could fail, leading to the failure of backdoor defenses built on this assumption. Specifically, we provided our insights on the phenomenon of latent separation, and designed adaptive attacks that can mitigate this separation. Empirical study and evaluation of various latent separation-based defenses showed that our adaptive poisoning attacks indeed suppress the latent separation and render them ineffective. We call for defense designers to take caution when leveraging latent separability as an assumption in their defenses. We also encourage further defenses to take our attacks into consideration for a more comprehensive evaluation.

ETHICS STATEMENT

During our study, we restricted all of our adversarial experiments within the laboratory environment, and did not induce any negative impact in the real world. Though our attacks may not be mitigated by many existing defenses, the illustration of our attacks is only conceptual. We encourage future work to design stronger defenses that resist our attacks.

ACKNOWLEDGEMENTS

This work was supported in part by the National Science Foundation under grants CNS-1553437 and CNS-1704105, the ARL's Army Artificial Intelligence Innovation Institute (A2I2), the Office of Naval Research Young Investigator Award, the Army Research Office Young Investigator Prize, Schmidt DataX award, Princeton E-ffiliates Award, and Princeton's Gordon Y. S. Wu Fellowship. Any opinions, findings, and conclusions or recommendations expressed in this material are those of the author(s) and do not necessarily reflect the views of the funding agencies.

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

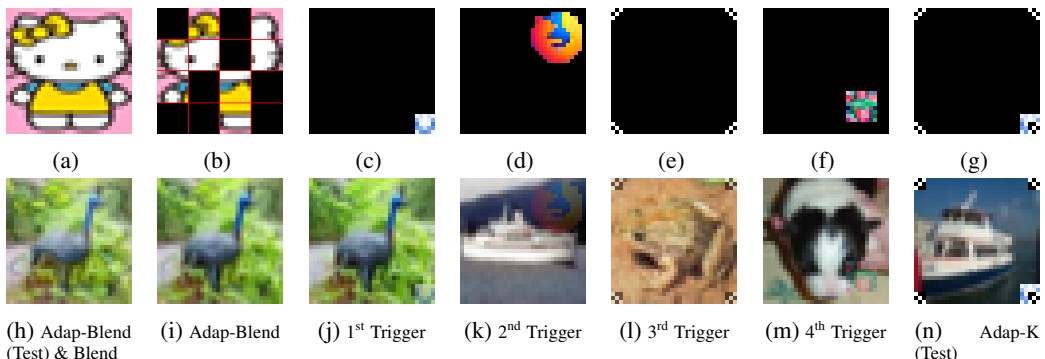

Figure 6: The example of poisoned samples.

# A    EXPERIMENT CONFIGURATIONS

## A.1    COMPUTATIONAL ENVIRONMENTS

All of our experiments are conducted on a workstation with 48 Intel Xeon Silver 4214 CPU cores, 384 GB RAM, and 8 GeForce RTX 2080 Ti GPUs.

## A.2    ATTACK CONFIGURATIONS

For each attack, we reuse the same triggers adopted in the paper. "Blend" and "Adap-Blend" use the blending-trigger (Fig 6a) with 20% opacity. "TaCT" uses the trojan square trigger (Fig6m) with 100% opacity. Eventually, "Adap-Patch" use the four triggers in Fig 6c-6f with 50%, 20%, 50% and 30% opacities, respectively. Specifically, each of their poison training sample randomly selects one of the four triggers (see Fig 6j-6m), while each test sample uses the enhanced two triggers demonstrated in Fig 6n.

The target class is set to class 0. 150 poison samples are used for "Blend", "BadNet", "ISSBA", "Dynamic" and "CL". For "TaCT", we use 150 poison samples and 150 cover samples. For "Adap-Blend", we use 150 payload samples and 150 regularization samples. For our "Adap-Patch", we use 150 payload samples and 300 regularization samples.

For all backdoor models, we adopt the standard training pipeline. SGD with a momentum of 0.9, a weight decay of $10^{-4}$, and a batch size of 128, is used for optimization. Initially, we set the learning rate to 0.1. On CIFAR-10, we follow the standard 200 epochs stochastic gradient descent procedure, and the learning rate will be multiplied by a factor of 0.1 at the epochs of 100 and 150. On GTSRB we use 100 epochs of training, and the learning rate is multiplied by 0.1 at the epochs of 40 and 80.

We consider widely adopted data augmentations, i.e. `RandomHorizontalFlip` and `RandomCrop` for CIFAR-10 and `RandomRotation` for GTSRB. We notice that data augmentation may affect defense results significantly, while defenders do not know whether to use augmentation or not. Therefore, we report the better defense result of the two backdoor models with and without augmentation. In another sentence, we report **upper-bound results** of the defenses which might be affected by the incorporation of data augmentation during backdoor training. Also, to rule out the effect of randomness, we train three models on three seeds for each configuration and report their average results.

### A.2.1    FAIRNESS CONSIDERATIONS

**Trigger Selections.**    For other non-adaptive attacks, we follow the trigger selections adopted in their original works in order to objectively evaluate how they perform against the defenses. For our attack, we argue that our trigger design itself is a vital part of our design — our triggers should not necessarily be the same to triggers of other attacks. Still, when we design Adap-Blend, we use the same trigger from Blend. The comparison between Blend and Adapt-Blend fairly reflects the effectiveness of our adaptive strategies. As for Adaptive-Patch, since it's difficult to partition the

Table 4: Additional results of comparing K-trigger and Adap-Patch attacks. Configurations are similar to Tab 1.

| Defenses→ | No Defense | | Spectral Signature | | Activation Clustering | | SCAn | | SPECTRE | |
|---|---|---|---|---|---|---|---|---|---|---|
| Attacks ↓ | ASR | CA | Elimination | Sacrifice | Elimination | Sacrifice | Elimination | Sacrifice | Elimination | Sacrifice |
| K-trigger | 100.0 | 91.5 | 62.7 | 4.3 | 0.0 | 3.5 | 92.0 | 11.5 | 94.7 | 0.2 |
| Adap-Patch | 98.8 | 91.0 | 10.0 | 4.5 | 0.0 | 0.0 | 0.0 | 0.0 | 0.0 | 0.5 |

Table 5: Additional results of non-adaptive attacks with 300 poison samples in total. Configurations are similar to Tab 1.

| Defenses→ | No Defense | | Spectral Signature | | Activation Clustering | | SCAn | | SPECTRE | |
|---|---|---|---|---|---|---|---|---|---|---|
| Attacks ↓ | ASR | CA | Elimination | Sacrifice | Elimination | Sacrifice | Elimination | Sacrifice | Elimination | Sacrifice |
| Blend | 95.3 | 91.8 | 67.0 | 8.6 | 85.3 | 16.1 | 94.7 | 0.0 | 98.7 | 0.3 |
| BadNet | 100.0 | 91.8 | 100.0 | 8.5 | 100.0 | 0.0 | 100.0 | 0.0 | 100.0 | 0.3 |
| ISSBA | 98.7 | 91.7 | 19.3 | 8.9 | 0.0 | 0.0 | 97.0 | 0.0 | 99.7 | 0.3 |
| Dynamic | 98.9 | 91.5 | 72.3 | 8.6 | 95.7 | 3.1 | 96.3 | 0.0 | 100.0 | 0.3 |
| CL | 99.6 | 92.0 | 100.0 | 8.5 | 100.0 | 0.1 | 100.0 | 0.0 | 100.0 | 0.3 |
| TaCT | 96.5 | 91.8 | 62.9 | 4.3 | 33.1 | 5.4 | 100.0 | 4.9 | 100.0 | 0.2 |

BadNet trigger further, instead, we turn to another set of patch triggers, which might be a potential unfair factor. To further address this fairness concern, hereby we provide in Tab 4 an extra set of controlled experiment (the poison rate follows the configurations used for Tab 1 in our paper): `K-trigger` attack that uses the same set of triggers as Adap-Patch for poisoning and testing, only without our adaptive regularization. Like Blend vs. Adap-Blend, K-trigger can thus be deemed as a non-adaptive counterpart to Adaptive-patch. As shown, K-trigger can still be eliminated by most of the latent-space defenses.

**Poison Rates.** We do not consider the (correctly labeled) regularization samples "malicious in our context", since they could not inject backdoor independently and do not induce latent separation. Only payload poison samples mislabelled to the target class are considered to be "malicious in our context". On the other hand, the stealthiness in the latent space is also strongly correlated with the number of payload samples — more malicious payload samples empirically lead to less stealthiness in the latent representation space, and thus are easier to be defended. So in Table 1, all attacks use the same number of malicious payload poison samples (exactly 150) in the target class, meeting the criteria of fairness in the sense of stealthiness. But, indeed, since Adap-Blend also uses an additional 150 (non-malicious) poison samples for regularization, it uses 300 poison samples in total, more than that of other non-adaptive attacks — but if we also use 300 poison samples for those non-adaptive attacks, they would be even less stealthy in the latent representation space, which is also unfair.

Still, we may want to know, *what if the total number of poison samples (counting both payload and regularization samples) are set the same?* To address this concern, in Tab 5, we supplement an extra set of experiment results on non-adaptive attacks, where each attack inject exactly 300 poison samples in total (same as Adap-Blend). Obviously, all these non-adaptive attacks still fail against latent-space defenses.

## A.3 DEFENSE CONFIGURATIONS

- Spectral Signature (Tran et al., 2018) removes $1.5 \cdot \rho_p$ suspected samples from every class.
- Activation Clustering (Chen et al., 2019) removes clusters with size <35% of the class size.
- SCAn (Tang et al., 2021) cleanses classes with scores larger than $e$.
- SPECTRE (Hayase et al., 2021) removes $1.5 \cdot \rho_p$ suspected samples only from the class with the highest QUE score.

## B MORE ABLATION STUDY

Since GTSRB and Imagenette contain different number of samples. For simplicity, we use the ratio of the whole training set to denote the number of poison samples, *i.e.*, poison rate. We use $\rho$ to

Table 6: Results of our adaptive attacks on GTSRB.

| Defenses→ | No Defense | | Spectral Signature | | Activation Clustering | | SCAn | | SPECTRE | |
|---|---|---|---|---|---|---|---|---|---|---|
| Attacks ↓ | ASR | CA | Elimination | Sacrifice | Elimination | Sacrifice | Elimination | Sacrifice | Elimination | Sacrifice |
| Dynamic | 100.0 | 98.0 | 96.2 | 17.8 | 75.9 | 3.4 | 84.8 | 3.2 | 0.0 | 0.1 |
| TaCT | 100.0 | 97.7 | 97.5 | 17.8 | 0.0 | 3.0 | 0.0 | 0.0 | 0.0 | 0.1 |
| Blend | 86.4 | 97.6 | 70.9 | 17.8 | 84.8 | 6.9 | 0.0 | 3.6 | 0.0 | 0.1 |
| **Adap-Blend** | 82.2 | 97.8 | 43.0 | 17.9 | 0.0 | 3.5 | 0.0 | 2.3 | 0.0 | 0.1 |
| BadNet | 99.5 | 97.8 | 100.0 | 25.2 | 97.7 | 2.3 | 99.2 | 0.5 | 100.0 | 0.3 |
| **Adap-Patch** | 63.1 | 97.8 | 74.4 | 25.4 | 0.0 | 0.3 | 35.6 | 4.6 | 0.0 | 0.1 |

denote the total ratio of poison samples. For our adaptive attacks, we use $\rho_p$ to denote the ratio of payload poison samples and $\rho_c$ to denote the ratio of regularization samples. Basically, $\rho = \rho_p + \rho_c$.

## B.1 ADAPTIVE ATTACKS ON DIFFERENT DATASETS

In this subsection, we evaluate our attacks on different datasets. The same ResNet-20 architecture that we use in the main experiment is consistently used for all the experiments in this subsection.

### B.1.1 GTSRB

GTSRB (Stallkamp et al., 2012) is a widely adopted dataset for backdoor study, for which we also evaluate our adaptive poisoning backdoor attacks on GTSRB.

Due to the imbalanced nature of GTSRB and the rotation-based data augmentation, we activate Adap-Patch with another set of asymmetric triggers, $i.e.$, Fig 6d and Fig 6f, for high ASR. We use $\rho_p = 0.003$ and $\rho_c = 0.003$ for "Adap-Blend". We notice that "Adap-Patch" has a lower ASR on GTSRB than on CIFAR-10 (due to more classes); to boost up the ASR, we use $\rho_p = 0.005$ and $\rho_c = 0.01$ for "Adap-Patch". For comparison, we also show results of "BadNet" ($\rho = 0.005$), "Blend" ($\rho = 0.003$), "Dynamic" ($\rho = 0.003$) and "TaCT" ($\rho_c = 0.003$ and $\rho_p = 0.003$).

As Table 6 tells, considerable amount of our adaptive poison samples could still survive the latent separability based defenses on GTSRB, as on CIFAR-10. Furthermore, compared with the non-adaptive attacks, our adaptive attacks consistently perform better (smaller elimination rates).

### B.1.2 IMAGENETTE

To illustrate the effectiveness adaptive strategy on high-resolution inputs (e.g., 224x224), we evaluate our adaptive poisoning backdoor attacks on a 10-classes Imagenet (Russakovsky et al., 2015) subset. Specifically, we take the commonly used Imagenette (Fastai, 2019) subset.

To successfully backdoor poison such a high-resolution dataset, selected triggers should usually be stronger (more evident visually). Specifically, for "Adap-Blend", we find the original blending trigger (Hellokitty) not strong enough, and replaced it with a random noise blending trigger (training poison alpha 0.15, test alpha 0.2; $\rho_p = 0.003$, $\rho_c = 0.003$); same trigger is used for "Blend" (alpha 0.2 all the time; $\rho_p = 0.003$). We also enhance "Adap-Patch" by training trigger opacities and their poison rates (Fig 6c-6d, opacity=1.0; $\rho_p = 0.01$, $\rho_c = 0.02$). For comparison, we use a stronger patch trigger ("Firefox", Fig6d) with $\rho_p = 0.01$ (the BadNet trigger is not strong enough for backdoor injection).

Table 7 shows that our adaptive attacks successfully circumvent the latent separability based defenses. On Imagenette, some defenses (Activation Clustering and SCAn with standard hyperparameters) are ineffective – they cannot cleanse any poison samples even for Blend and Firefox. Even so, we can still notice the superiority of our adaptive strategy over naive poisoning attacks (significantly smaller elimination rates against Spectral Signature and SPECTRE).

## B.2 ADAPTIVE ATTACKS ON DIFFERENT ARCHITECTURES

We also consider other model architectures, e.g., VGG-16 (Simonyan & Zisserman, 2014) and MobileNet-V2 (Sandler et al., 2018). We evaluate these architectures on CIFAR-10 against our

Table 7: Results of our adaptive attacks on Imagenette.

| Defenses→ | No Defense | | Spectral Signature | | Activation Clustering | | SCAn | | SPECTRE | |
|---|---|---|---|---|---|---|---|---|---|---|
| Attacks ↓ | ASR | CA | Elimination | Sacrifice | Elimination | Sacrifice | Elimination | Sacrifice | Elimination | Sacrifice |
| Blend | 93.1 | 90.1 | 21.4 | 4.4 | 0.0 | 3.2 | 0.0 | 0.0 | 100.0 | 0.1 |
| Adap-Blend | 60.8 | 89.7 | 7.1 | 4.4 | 0.0 | 3.1 | 0.0 | 0.0 | 0.0 | 0.4 |
| Firefox | 98.3 | 89.8 | 70.2 | 14.4 | 0.0 | 2.4 | 0.0 | 0.0 | 100.0 | 9.9 |
| Adap-Patch | 85.7 | 89.5 | 14.9 | 15.0 | 0.0 | 2.5 | 0.0 | 2.7 | 0.0 | 1.5 |

Table 8: Results of our adaptive attacks on other network architectures.

| Defenses↓ | Archs→ | VGG-16 | | MobileNet-V2 | |
|---|---|---|---|---|---|
| | Attacks→ | Adap-Blend | Adap-Patch | Adap-Blend | Adap-Patch |
| Without Defense | ASR | 74.0 | 63.6 | 72.2 | 96.4 |
| | Clean Accuracy | 93.4 | 93.6 | 92.2 | 92.3 |
| Spectral Signature | Elimination Rate | 59.3 | 53.3 | 3.3 | 2.7 |
| | Sacrifice Rate | 4.3 | 4.4 | 4.5 | 4.5 |
| Activation Clustering | Elimination Rate | 0.0 | 0.0 | 0.0 | 0.0 |
| | Sacrifice Rate | 0.0 | 0.0 | 0.0 | 0.0 |
| SCAn | Elimination Rate | 0.0 | 0.0 | 0.0 | 0.0 |
| | Sacrifice Rate | 0.0 | 0.0 | 0.0 | 4.7 |
| SPECTRE | Elimination Rate | 0.0 | 0.0 | 0.0 | 0.0 |
| | Sacrifice Rate | 0.5 | 0.5 | 0.5 | 0.5 |

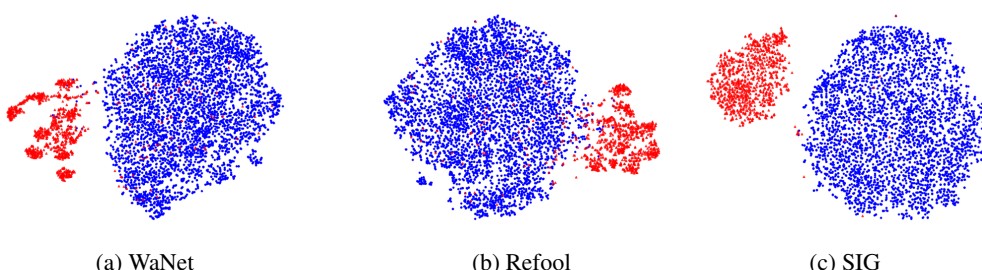

(a) WaNet      (b) Refool      (c) SIG

Figure 7: t-SNE Visualization of latent separability of WaNet, Refool and SIG on CIFAR-10.

adaptive attacks. Notice that these architectures have much larger latent spaces – MobileNet-V2 has a latent space of dimension 1280, and VGG-16 has a latent space of dimension 512 (for comparison, 64 for ResNet-20). Therefore, we have to adapt some defense configurations. Specifically, SCAn could not finish computing for MobileNet-V2 after 2h, so we manually reduce the latent representation's dimension to 128 by PCA before SCAn starts processing.

Overall, we show that our adaptive attacks also effectively circumvent latent separation based defenses. In Table 8, we demonstrate our adaptive attacks with these network architectures on CIFAR-10. As shown, none of these latent separability based defenses could completely eliminate our backdoor poison samples and most of them completely fail.

## C OTHER STATE-OF-THE-ART BACKDOOR ATTACKS

We also consider some other state-of-the-art backdoor attacks built on "advanced triggers". We illustrate that they also suffer from latent separation. We take Refool (Liu et al., 2020), WaNet (Nguyen & Tran, 2021) and SIG (Barni et al., 2019) as examples. Refool constructs realistic reflection triggers for backdoor injection, WaNet uses warp transformation for each image as the backdoor trigger, and SIG uses superimposed sinusoidal signals as triggers. We implement them in a poison-only manner, where they require much larger poison rates ($\rho = 0.02$ for all three attacks) to inject backdoors with non-trivial ASR. Their T-SNE visualizations in Fig 7 clearly reveals the latent separation.

Table 9: The resistance of our adaptive attacks to other types of defenses on CIFAR-10. "ASR" for attack success rate, "CA" for clean accuracy, "Eli" for elimination rate, "Sac" for sacrifice rate, "IP" for isolation precision, "STRIP (C) & (F)" for STRIP as a poison cleanser and an input filter.

| Defenses→ | FP | | STRIP (C) | | STRIP (F) | | NC | | ABL | NAD | |
|---|---|---|---|---|---|---|---|---|---|---|---|
| Attacks ↓ | ASR | CA | Eli | Sac | Eli | Sac | ASR | CA | IP | ASR | CA |
| Blend | 78.1 | 81.1 | 17.3 | 9.7 | 14.3 | 10.0 | 87.5 | 91.9 | 0.4 | 4.9 | 80.8 |
| **Adap-Blend** | 77.5 | 76.9 | 0.7 | 9.7 | 7.4 | 10.0 | 72.4 | 91.5 | 0.0 | 9.5 | 81.2 |
| BadNet | 88.9 | 80.7 | 100.0 | 10.1 | 100.0 | 10.0 | 1.7 | 90.3 | 4.2 | 3.1 | 83.1 |
| **Adap-Patch** | 99.5 | 80.9 | 21.3 | 10.6 | 99.9 | 10.0 | 2.2 | 89.2 | 0.0 | 19.5 | 81.3 |

## D  OTHER DEFENSES

Beyond the latent-space defenses, there are also other types of backdoor defenses (e.g., (Liu et al., 2018; Gao et al., 2019; Wang et al., 2019; Li et al., 2021a;b; Borgnia et al., 2021a; Li et al., 2021c; Borgnia et al., 2021b; Shen et al., 2021)) that are not explicitly based on the latent separability assumption. These defenses could potentially ease the threat brought by our adaptive attacks. For a more comprehensive evaluation, we also examine our adaptive attacks against (some of) them. Specifically, we consider:

- Fine-Pruning (FP) (Liu et al., 2018): a *model-pruning-based* backdoor defense that claims when a model is fed with clean inputs, its dormant neurons are more likely to be responsible for the backdoor task. FP eliminates a model's backdoor by pruning these dormant neurons until a certain clean accuracy drop.

- STRIP (Gao et al., 2019): an input-filtering-based backdoor defense based on the observation that when a poison sample is superimposed by clean samples, the predicted class confidence drops heavily. We consider STRIP both as a cleanser (STRIP (C) detects poison training samples) and a test-time filter (STRIP (F) detects poison test samples).

- Neural Cleanse (NC) (Wang et al., 2019): a *reverse-engineering-based* backdoor defense that restores triggers by optimizing on the input domain. The authors claim a class with its reversed trigger having an abnormally small norm is more possibly a poisoned target class. Quantitatively, it calculates an anomaly index for each class w.r.t. the reversed triggers' mask norm, where classes with anomaly index ¿2 are judged as poisoned targets (outliers). Then, the smallest abnormal reversed trigger is patched on a small clean set to unlearn the model's backdoor.

- Anti-Backdoor Learning (ABL) (Li et al., 2021a): a *poison-suppression-based* backdoor defense that utilizes local gradient ascent to isolate 1% suspected training samples with the smallest losses. The authors claim these samples are more possible to be poison samples and may help unlearn the backdoor.

- Neural Attention Distillation (NAD) (Li et al., 2021b): a *distillation-based* backdoor defense that unlearn backdoor in a teacher-student distillation fashion. NAD first finetunes the backdoor model with a holdout clean set to obtain a "teacher" model, then uses this "teacher" model to teach the original backdoor model ("student") to further unlearn the backdoor.

As shown in Table 9: our adaptive methods do not decrease the resistance of their vanilla version to defenses other than latent-space ones. For STRIP (C), our adaptive attacks even significantly outperform their vanilla version – possibly due to the weaker correlation between the backdoor trigger and the target label in our poison samples. We also show in Table 10 and Figure 8 that by selecting different test-time triggers, an Adap-Patch attacker can further evade the test-time STRIP (F) backdoor filter.

## E  QUANTIFICATION OF LATENT (IN)SEPARABILITY

**Silhouette Score (Coefficient)** we show in Table 1 formally measures such a property. For our case, Silhouette Score (Coefficient) measures how different are two labeled clusters (*i.e.*, knowing

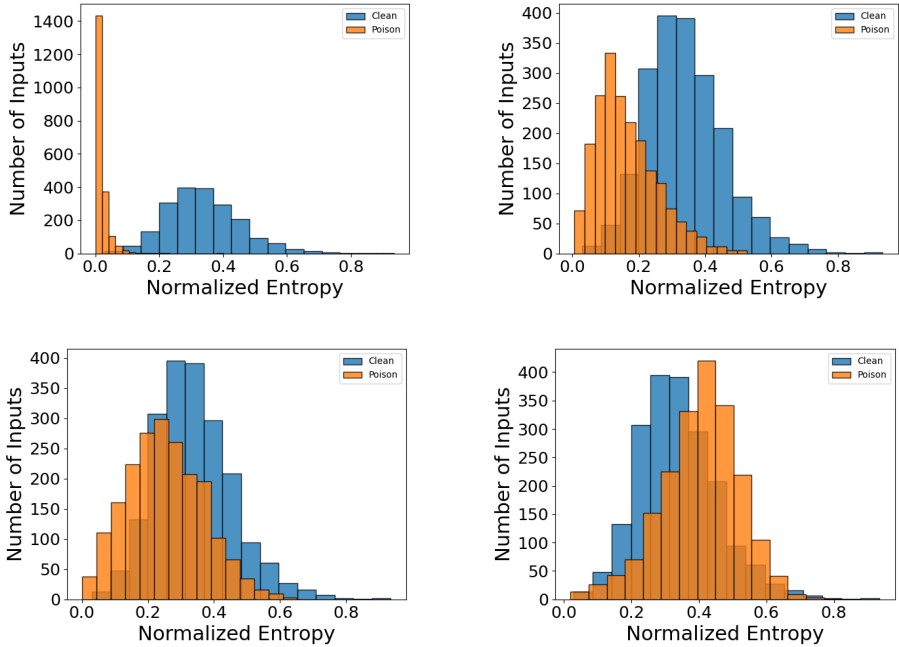

Figure 8: The STRIP normalized entropy histograms of Adap-Patch with different trigger selections. Samples with smaller entropy are more suspected to be poisoned. As shown, an attacker can achieve tradeoff between ASR and stealthiness by choosing different test-time triggers. Please refer to Table 10 for details of defense results.

Table 10: STRIP (as an input filter) against Adap-Patch attack with different trigger selections. The corresponding normalized histograms are shown in Fig 8.

| Triggers | ASR | Eli | Sac |
|---|---|---|---|
| Trigger 6c + Trigger 6e | 97.5 | 99.0 | 10.0 |
| Trigger 6d + Trigger 6m | 86.5 | 70.0 | 10.0 |
| Trigger 6c (opacity=0.5) + Trigger 6e (opacity=0.7) | 59.7 | 35.0 | 10.0 |
| Trigger 6d | 25.6 | 5.3 | 10.0 |

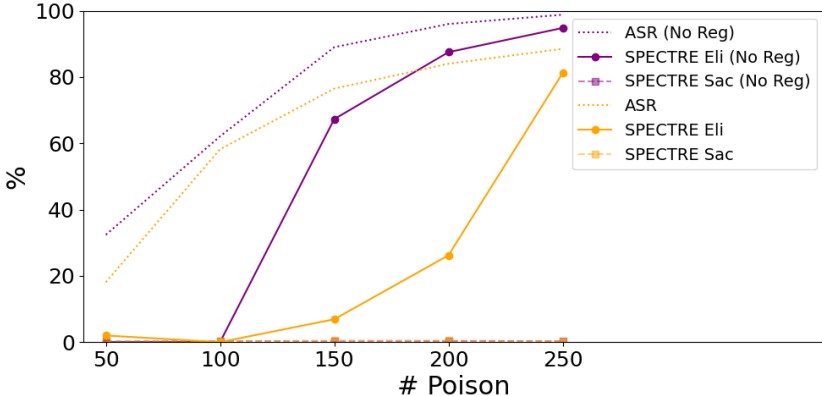

Figure 9: Results of Adap-Blend with (orange) & without (purple) regularization samples. The dotted lines show the ASR. The solid lines correspond to Elimination ("Eli") and the dotted lines correspond to Sacrifice ("Sac"). As shown, Adap-Blend with regularization could remain stealthier in a larger range of poison rate selections (elimination $< 30\%$ for 0-200 payload samples), compared to Adap-Blend without regularization that only uses our asymmetric triggers (elimination $> 65\%$ for 150+ payload samples). **This further confirms the necessity of regularization.**

which samples are poisoned). The Silhouette Coefficient is calculated using the mean intra-cluster distance ($a$) and the mean nearest-cluster distance ($b$) for each sample. The Silhouette Coefficient for a sample is $(b - a)/\max(a, b)$. ($b$ is the distance between a sample and the nearest cluster that the sample is not a part of; $i.e.$, the other cluster in our case.) As shown in the last row of Table 1, our Adap-attacks have significantly lower Silhouette Scores, which means the clean and poison clusters overlap more (and are thus more indistinguishable).

## F   THE COMPARISON WITH LSBA

We notice a concurrent work (Peng et al., 2022) proposing Label-Smoothed Backdoor Attack (LSBA) that shares certain similarity with ours adaptive backdoor attacks. We here list several fundamental differences between our work and LSBA: **1) Motivation**. Our method is motivated by our observations that the latent separability is so pervasive across both classical and advanced backdoor poisoning attacks. Peng et al. (2022) was inspired by the overfitting problem of existing backdoor attacks. **2) The Generation of Regularization/Poisoned Samples**. In general, both our method and Peng et al. (2022) preserve the ground-truth label (instead of re-assigning to the target label) of a fraction of modified samples. However, the fraction of our method is sample-independent whereas that of Peng et al. (2022) is sample-specific. This difference is because we have different mechanisms: we intend to alleviate the latent separability whereas they aimed to assign a soft-label to each poisoned sample. **3) The Improvement Module**. In general, both our method and Peng et al. (2022) have an additional module to further improve the main attack. However, we adopt asymmetric triggers to alleviate latent separability and encourage trigger diversity while preserving high attack effectiveness, whereas Peng et al. (2022) implanted multiple different backdoors to improve attack effectiveness. **4) Baseline Defenses**. We evaluate our attacks against latent separation based defenses, because our goal is to design adaptive backdoor poisoning attacks against this family of defenses. In contrast, Peng et al. (2022) only adopted STRIP (Gao et al., 2019) and Neural Cleanse (Wang et al., 2019) for evaluation. These two defenses are not based on latent separation.

Moreover, our method has unique contributions:

1. We formulate and reveal the latent separability assumption that is underlying many state-of-the-art defenses. We demonstrate that the latent separability assumption holds across a diverse set of backdoor poisoning attacks in the existing literature. Our work highlights the big blank in designing backdoor poisoning attacks that are stealthy in the latent representation space, and might consequently motivate future research in this direction.

2. In this work, we have done a proof-of-concept study to show that the latent separability assumption could fail. This conclusion suggests defense designers take caution when leveraging latent separation as an assumption in their defenses. We believe this can influence the design of backdoor defenses in the future.

## G   THE COMPARISON WITH TACT

TaCT (Tang et al., 2021) also uses a regularization technique. Specifically, TaCT injects regularization samples from several "cover classes" to ensure that only images from a "source class" are able to trigger the model's backdoor behavior when planted with the backdoor trigger. Whilst, our adaptive attack does not presume such "source class", and our regularization strategy is to randomly select regularization samples from all classes.

We here provide two extra ablation experiments to show that TaCT can hardly evade latent-space defenses, even when: **1)** injecting fewer poison samples (we explore the cases when there are only 50, 100, and 150 payload samples), and **2)** adopting our asymmetric trigger design (we use the same $k = 4$ triggers from Adap-Patch following TaCT's regularization labelling strategy, denoted as `TaCT-k`). We show the defense results by SPECTRE in Tab 11.

Clearly, even with fewer poison samples and the asymmetric k-triggers, TaCT (TacT-k) cannot evade latent-space detection ($\geq 90\%$ payload samples are cleansed by SPECTRE). On the other hand, both our adaptive attacks can hardly be detected (only $< 7\%$ payload samples are cleansed by SPECTRE), and thus significantly outperform TaCT $w.r.t.$ stealthiness against latent space defenses.

Table 11: Results of TaCT and TaCT-k with fewer payload samples. We use $\rho_{\text{regularization}} = \rho_{\text{payload}}$ for TaCT. for TaCT-k, we use $\rho_{\text{regularization}} = 2 \cdot \rho_{\text{payload}}$. We only show defense results of SPECTRE since it consistently provides the highest poison elimination.

| Attack↓ | # Payload Samples → | 50 | 100 | 150 |
|---|---|---|---|---|
| TaCT | ASR (w/o defense) | 93.9 | 96.9 | 95.1 |
| | Clean Acc (w/o defense) | 91.8 | 91.5 | 91.7 |
| | SPECTRE Elimination Rate | 100.0 | 100.0 | 100.0 |
| | SPECTRE Sacrifice Rate | 0.1 | 0.1 | 0.2 |
| TaCT-k | ASR (w/o defense) | 95.4 | 100.0 | 100.0 |
| | Clean Acc (w/o defense) | 92.0 | 91.9 | 91.8 |
| | SPECTRE Elimination Rate | 90.0 | 95.0 | 99.3 |
| | SPECTRE Sacrifice Rate | 0.1 | 0.1 | 0.2 |
| **Adap-Blend** | ASR (w/o defense) | 18.1 | 58.2 | 76.5 |
| | Clean Acc (w/o defense) | 91.8 | 91.4 | 91.6 |
| | SPECTRE Elimination Rate | 2.0 | 0.0 | 6.9 |
| | SPECTRE Sacrifice Rate | 0.1 | 0.3 | 0.5 |
| **Adap-Patch** | ASR (w/o defense) | 75.1 | 83.0 | 97.5 |
| | Clean Acc (w/o defense) | 91.6 | 91.8 | 91.9 |
| | SPECTRE Elimination Rate | 6.0 | 0.0 | 0.0 |
| | SPECTRE Sacrifice Rate | 0.1 | 0.3 | 0.5 |

Table 12: Additional results of comparing M-Way and Adap-M-Way attacks.

| Defenses↓ | Attacks→ | M-Way | Adap-M-Way |
|---|---|---|---|
| Without Defense | ASR | 87.7 | 73.8 |
| | Clean Accuracy | 91.3 | 91.0 |
| Spectral Signature | Elimination Rate | 33.3 | 19.0 |
| | Sacrifice Rate | 4.4 | 4.5 |
| | ASR | 4.7 | 66.6 |
| | Clean Accuracy | 91.3 | 90.9 |
| Activation Clustering | Elimination Rate | 0.0 | 0.0 |
| | Sacrifice Rate | 0.0 | 0.0 |
| | ASR | 87.8 | 73.8 |
| | Clean Accuracy | 91.3 | 91.0 |
| SCAn | Elimination Rate | 55.3 | 0.0 |
| | Sacrifice Rate | 7.5 | 0.0 |
| | ASR | 3.3 | 73.8 |
| | Clean Accuracy | 91.0 | 91.0 |
| SPECTRE | Elimination Rate | 64.0 | 10.0 |
| | Sacrifice Rate | 0.3 | 0.4 |
| | ASR | 2.4 | 62.5 |
| | Clean Accuracy | 91.4 | 91.5 |

# H   THE COMPARISON WITH M-WAY ATTACK

Xie et al. (2019) proposes M-Way attack as an instance of distributed backdoor attack in federated learning scenarios, where $m$ independent pixels are used as $m$ distributed backdoor triggers, respectively. They proposes DBA targeting federated learning scenarios, and also studies a type of asymmetric triggers for data poisoning and testing time attacks respectively. Compared with our work, (1) M-Way attack still suffers from latent separation; (2) it doesn't consider regularization in data poisoning as well as its implication for suppressing latent separation characteristics.

To highlight the differences, we:

1. Supplement an additional evaluation on the M-Way attack on CIFAR10. We follow the original settings of Xie et al. (2019), and use the same number of payload samples to that of our main evaluation in Table 1.

2. Further implement an adaptive version of the M-Way attack (Adap-M-Way) via incorporating our idea of regularization samples. Number of payload and regularization samples are also the same to that of Adaptive-Blend in Table 1.

As shown in Table 12, the original M-Way attack proposed by Xie et al. (2019) can be defended (ASR< 5%) by 3 out of 4 latent separation-based defenses that we evaluate against. After being adapted to our regularization techniques, none of the four attacks can defend it.

