# OpenReview forum: "Revisiting the Assumption of Latent Separability for Backdoor Defenses"
_ICLR.cc/2023/Conference — ICLR 2023 poster_

### Official Review · Reviewer_YTpa · 2022-10-23

**Confidence:** 4
**Correctness:** 3
**Technical Novelty And Significance:** 3
**Empirical Novelty And Significance:** 3
**Recommendation:** 8

**Clarity, Quality, Novelty And Reproducibility:**

There has been decent amount of work on making backdoor poisoning stealthy, but the specific formulation this paper offers is original. The evaluation is done thoroughly but I left some feedback down below.

**Strength And Weaknesses:**

Strengths:

+ Interesting formulation, the idea of weak triggers combining into a single trigger is novel and seems to be effective to make the attack stealthy.

+ Most work on making backdoor attacks stealthy focuses on supply-chain attacks (not poisoning). This formulation can enable further research in a more realistic threat model.

+ Well written and evaluated on recent attacks and defenses from multiple perspectives to show the benefits of the proposed attack.

Weaknesses:

- The comparisons between baseline attacks and the proposed attack might be unfair and difficult to interpret.

- Ablation study needs a little more work.

**Summary Of The Paper:**

This paper proposes a backdoor poisoning attack that aims to defeat latent separability-based defenses that purify the training set. The attack has two main components: regularization samples that contain the trigger but assigned to the correct class and asymmetric triggers that are weaker versions of the actual trigger planted into the poison samples.  As a result, the attack avoids recent defenses more successfully than baseline evasive attacks.

**Summary Of The Review:**

1) My biggest concern is that the baseline attacks and the proposed attack have some configuration differences that make comparisons difficult/unfair.

For example, TaCT uses the trigger in Fig 6m at test-time, an NxN trigger. On the other hand, the proposed attack uses two different triggers overlaid on the image at the test time (the corner pattern and a square). Further, some attacks inject 150 samples and others inject 300 samples (TaCT and the proposed attack). This again makes the threat models inconsistent.

I recommend the authors to first clearly specify the treat model and the attacker's capabilities across the board, e.g., P poison samples [patch] test-time: NxN trigger and training-time: NxN or smaller a trigger or [blend] maximum opacity levels at the training and testing times. Then, run all the attacks within these constraints to make sure that the same attacker can perform any of these attacks within their constraints. Right now, it's not clear if that's the case for all attacks as the threat models in each have minor differences.

2) When evaluating the defenses, the default hyper-parameters have different impact on each attack. For example, SCAn sacrifice rate for TaCT is 4.9% whereas it is 1.2% for Adapt-Blend. This means, SCAn is less conservative on TaCT, which might explain the high elimination rate. I would recommend tuning the defenses across the board to achieve the same sacrifice rate (e.g., 1% - 2% - 5%) and then reporting the elimination rate, which would make the comparisons more fair. Alternatively, you can put a hard threshold over the clean accuracy drop caused by the defense (e.g., 0.1% - 0.5% etc.) then report the corresponding elimination/sacrifice rates.

3) Ablation study doesn't show that regularization is absolutely necessary. Would it be possible to achieve the same impact by reducing the number of poisons (150 - 100 - 50 - 25 etc.) while not adding any regularization sample? In Table 2, you experimented with 150 poison- 0 regularization but what about 100 poison - 0 reg. for example?

4) I know the point of the paper is to evade latent clustering-based defenses but what about other defenses? For example, is the proposed attack more effective against trigger reverse-engineering [1], distillation [2] or data augmentation [3,4]? It would be great to have a small section on these different defensive methods as well. My worry is that the weak triggers would make the attack easier to defuse especially using [2,3,4], would be great to show otherwise.

[1] https://arxiv.org/abs/2102.05123
[2] https://openreview.net/forum?id=9l0K4OM-oXE
[3] https://arxiv.org/abs/2011.09527
[4] https://arxiv.org/pdf/2103.02079.pdf

---

> ### Author Response · Authors · 2022-11-14
> **Author Response (YTpa) --- Part I**
>
> We sincerely thank you for your valuable time and comments. We are encouraged by your positive comments on our **formulation**, **novelty**, **valuable contributions**, and **paper writing**! We will alleviate your remaining concerns in this rebuttal, as follows.
>
> ---
>
> **Q1:** Configuration differences that make comparisons difficult/unfair.
>
> **A1:** We thank the reviewer for raising the "fairness" issue in backdoor attack evaluations. We first want to point out that --- it is intrinsically challenging to keep absolutely the same configurations for all different backdoor attacks during comparison. For example, trigger design is oftentimes a part of attack design itself. Different attacks often use their own unique triggers that are not necessarily designed w.r.t. the same perturbation bound. Badnet[1] uses a patch as the trigger. Blend [2] uses global image blending. Dynamic[3] uses sample-specific dotted points. ISSBA[4] uses image steganography... **Nevertheless, we still try our best to achieve the fairness goal when reporting our results**. Here, we explain our efforts for a fair comparison as well as supplement more results that hopefully would alleviate the reviewer's concerns.
> 1.  The first part of the reviewer's concerns is about **trigger selection**. For other attacks, we follow the trigger selections adopted in their original works in order to objectively evaluate how they perform against the defenses. For our attack, we argue that **our trigger design itself is a vital part of our design** --- our triggers should not necessarily be the same to triggers of other attacks. Still, when we design Adap-Blend, we use **the same trigger** from Blend. The comparison between Blend and Adapt-Blend fairly reflects the effectiveness of our adaptive strategies. As for Adaptive-Patch, since it's difficult to further partition the BadNet trigger, indeed, we turn to another set of patch triggers, which might thus make the reviewer concern. To further address the reviewer's concerns, hereby we provide an extra set of controlled experiment (the poison rate follows the configurations used for Table 1 in our paper): *K-trigger* that uses the same set of triggers as Adap-Patch for poisoning and testing, only without our adaptive regularization. Like Blend vs. Adap-Blend,  **K-trigger can thus be deemed as a non-adaptive counterpart** to Adaptive-patch.
>     | Defenses$\rightarrow$ | No Defense | SS | AC | SCAn | SPECTRE |
>     | -------- | -------- | -------- | -------- | -------- | -------- |
>     | **Attacks $\downarrow$**     | **ASR (CA)** | **ER (SR)**  | **ER (SR)**  | **ER (SR)**  |     **ER (SR)** |
>     | K-trigger | 100.0 (91.5) | 62.7 (4.3) | 0.0 (3.5) | 92.0 (11.5) | 94.7 (0.2) |
>     | **Adap-Patch** | 98.8 (91.0) | 10.0 (4.5) | 0.0 (0.0) | 0.0 (0.0) | 0.0 (0.5) |
>
>
> ("ASR" for attack success rate, "CA" for clean accuracy, "ER" for elimination rate, "SR" for sacrifice rate. We follow the same poison rate settings in our main experiment.)

---

> > ### Author Response · Authors · 2022-11-14
> > **Author Response (YTpa) --- Part II**
> >
> > **(Following A1 above)**
> >
> > 2. The other part of the reviewer's concerns is about poison rates. Actually, we have already carefully considered such an issue when reporting our results.
> >     - In our paper, we do not consider the (correctly labeled) regularization samples malicious, since they could not inject backdoor independently. Only payload poison samples mislabelled to the target class are considered to be malicious. On the other hand, the stealthiness in the latent space is also strongly correlated with the number of payload samples --- more malicious payload samples empirically lead to less stealthiness in the latent representation space, and thus are easier to be defended. So in Table 1 of our paper, all attacks use the same number of malicious payload poison samples (exactly 150) **in the target class**, meeting the criteria of fairness in the sense of stealthiness. But, indeed, since Adap-Blend also uses an additional 150 (non-malicious) poison samples for regularization, it uses 300 poison samples in total, more than that of other non-adaptive attacks --- but if we also use 300 poison samples for those non-adaptive attacks, they would be even less stealthy in the latent representation space, which would also be unfair.
> >     - Still, we understand that the reviewer may want to know, what if the total number of poison samples (counting both payload and regularization samples) are set the same? To address this concern, below we supplement an extra set of experiment results on non-adaptive attacks, where each attack inject exactly 300 poison samples in total (same as Adap-Blend):
> >
> >         Table: Results of other attacks with 300 poison samples (the same as Adap-Blend).
> >
> >         | Defenses$\rightarrow$ | No Defense | SS | AC | SCAn | SPECTRE |
> >         | -------- | -------- | -------- | -------- | -------- | -------- |
> >         | **Attacks $\downarrow$**     | **ASR (CA)** | **ER (SR)**  | **ER (SR)**  | **ER (SR)**  | **ER (SR)** |
> >         | Blend | 95.3 (91.8) | 67.0 (8.6) | 85.3 (16.1) | 94.7 (0.0) | 98.7 (0.3) |
> >         | BadNet | 100.0 (91.8) | 100.0 (8.5) | 100.0 (0.0) | 100.0 (0.0) | 100.0 (0.3) |
> >         | ISSBA | 98.7 (91.6) | 19.3 (8.9) | 0.0 (0.0) | 97.7 (0.0) | 99.7 (0.3) |
> >         | Dynamic | 98.9 (91.5) | 72.3 (8.6) | 95.7 (3.1) | 96.3 (0.0) | 100.0 (0.3) |
> >         | CL | 99.6 (92.0) | 100.0 (8.5) | 100.0 (0.1) | 100.0 (0.0) | 100.0 (0.3) |
> >         | TaCT | 96.5 (91.8) | 62.9. (4.3) | 33.1 (5.4) | 100.0 (4.9) | 100.0 (0.2) |
> >
> >         ("ASR" for attack success rate, "CA" for clean accuracy, "ER" for elimination rate, "SR" for sacrifice rate.)
> >
> > We will also add an independent section in our appendix to further discuss this fairness issue.
> >
> > ---
> >
> > **Q2:** When evaluating the defenses, the default hyper-parameters have a different impact on each attack. For example, SCAn sacrifice rate for TaCT is 4.9% whereas it is 1.2% for Adapt-Blend. This means, SCAn is less conservative on TaCT, which might explain the high elimination rate. I would recommend tuning the defenses across the board to achieve the same sacrifice rate (e.g., 1% - 2% - 5%) and then reporting the elimination rate, which would make the comparisons more fair. Alternatively, you can put a hard threshold over the clean accuracy drop caused by the defense (e.g., 0.1% - 0.5% etc.) then report the corresponding elimination/sacrifice rates.
> >
> > **A2:** We thank the reviewer for suggesting additional control on the sacrifice rate. We agree that a calibrated sacrifice rate would make our results more comprehensive.
> > 1. Actually, two defenses (Spectral Signature and SPECTRE) are indeed well calibrated in our paper, because these defenses always deletes a fixed percentage of samples with highest outlier scores.
> > 2. However, the other two defenses (Activation Clustering and SCAn) can not be calibrated by design --- they directly perform unsupervised cluster analysis to divide samples into two clusters and then delete the smaller cluster (no guarantee on the size of each cluster, because it is unsupervised). To faithfully reimplement these prior arts, we directly follow their original implementations.

---

> > > ### Author Response · Authors · 2022-11-14
> > > **Author Response (YTpa) --- Part III**
> > >
> > > **Q3**: Ablation study doesn't show that regularization is absolutely necessary. Would it be possible to achieve the same impact by reducing the number of poisons (150 - 100 - 50 - 25 etc.) while not adding any regularization sample? In Table 2, you experimented with 150 poison- 0 regularization but what about 100 poison - 0 reg. for example?
> > >
> > > **A3**: Thank you for the careful consideration about the necessity of regularization.
> > > 1. Our results in Table-3a already show that simply reducing poison rate could not help non-adaptive blending attack to evade the defenses.
> > > 2. On the other hand, as we have mentioned in our paper, our asymmetric trigger design could also help to suppress the latent separation --- so, with even lower poison rate (50 or 100 payload poison samples) as suggested by the reviewer, merely incorporating the asymmetric trigger design (without regularization) is indeed sufficient to bypass existing latent space defenses (Fig 9 in Appendix). Nonetheless, the ASR could only reach $32.4\%$ and $62.1\%$, respectively (while our best implementation of Adap-Blend reaches an ASR of $76.5\%$).
> > > 3. Besides, we want to highlight that --- with the regularization technique, our adaptive poisoning attacks could remain **stealthier** in a larger range of poison rate selections (elimination $<30\%$ for 0-200 payload samples), compared to the attack that only use our asymmetric triggers (elimination $>65\%$ for 150+ payload samples). See our Fig 9 in Appendix for the comparison between our adaptive attack with \& without regularization of different poison rates. This further validates the necessity of regularization.
> > >
> > > ---
> > >
> > > **Q4**: Other defenses that are not based on latent separation.
> > >
> > > **A4**: We also thank the reviewer for the suggestion to evaluate beyond latent space defenses. Initially, We mainly evaluated our method under latent-space defenses since our main target is to circumvent the assumption of latent separability that was regarded as fundamental and essential for poison-only backdoor attacks in previous works. However, we do understand your concern that our method may make the attack more detectable against other defenses. To alleviate this concern, we conduct additional experiments (Tab 2), including trigger reverse-engineering, distillation, data perturbation/augmentation, pruning, and training-stage defenses. Specifically, we consider:
> > >
> > > 1. Fine Pruning (FP[5]), Neural Cleanse (NC[7]), Neural Attention Distillation (NAD[9]) that aim to remove the backdoor. We report Attack Sucess Rate (ASR) and Clean Accuracy (CA) after these defenses.
> > > 2. STRIP\(C\) and STRIP(F), the two variants of STRIP [6] for detecting backdoor training samples and filtering test time backdoor samples respectively. We report the elimination rate (ER) of backdoor poison samples and sacrifice rate of clean samples (SR).
> > > 3. Anti-backdoor Learning (ABL[8]) that isolates backdoor training samples based on the speed of fitting. We report its isolation precision (IP) used by the original paper.
> > >
> > > Table: The resistance to additional defenses other than latent-space-based ones.
> > > | Defenses$\rightarrow$ | FP [5] | STRIP \(C\) [6] | STRIP (F) [6] | NC [7] | ABL [8] | NAD [9] |
> > > | -------- | -------- | -------- | -------- | -------- | -------- | -------- |
> > > | **Attacks $\downarrow$**     | **ASR (CA)** | **ER (SR)**  | **ER (SR)**  | **ASR (CA)** | **IP**  | **ASR (CA)** |
> > > | Blend | 78.1 (81.1) | 17.3 (9.7) | 14.3 (10.0) | 87.5 (91.9) | 0.4 | 4.9 (80.8) |
> > > | **Adap-Blend** | 77.5 (76.9) | 0.7 (9.7) | 7.4 (10.0) | 72.4 (91.5) | 0.0 | 9.5 (81.2) |
> > > | BadNet | 88.9 (80.7) | 100.0 (10.1) | 100.0 (10.0) | 1.7 (90.3) | 4.2 | 3.1 (83.1) |
> > > | **Adap-Patch** | 99.5 (80.9) | 21.3 (10.6) | 99.9 (10.0) | 2.2 (89.2) | 0.0 | 19.5 (81.3) |
> > >
> > > ("ASR" for attack success rate, "CA" for clean accuracy, "ER" for elimination rate, "SR" for sacrifice rate, "IP" for isolation precision of poison samples, "STRIP \(C\) \& (F)" for STRIP as a poison cleanser and an input filter, respective.)
> > >
> > > The aforementioned results show that our adaptive methods do not decrease the resistance against other defenses that are not based on latent separation. Please refer to our Appendix D in revision for more details. In our final version, we will keep adding all other state-of-the-art defenses suggested by the reviwer for comprehensiveness.

---

> > > > ### Author Response · Authors · 2022-11-14
> > > > **Author Response (YTpa) --- Part IV**
> > > >
> > > > **References**
> > > >
> > > > [1] Tianyu Gu, Brendan Dolan-Gavitt, and Siddharth Garg. Badnets: Identifying vulnerabilities in the machine learning model supply chain. arXiv preprint arXiv:1708.06733, 2017.
> > > >
> > > > [2] Xinyun Chen, Chang Liu, Bo Li, Kimberly Lu, and Dawn Song. Targeted backdoor attacks on deep learning systems using data poisoning. arXiv preprint arXiv:1712.05526, 2017.
> > > >
> > > > [3] Tuan Anh Nguyen and Anh Tran. Input-aware dynamic backdoor attack. In NeurIPS, pp. 3454–3464, 2020
> > > >
> > > > [4] Yuezun Li, Yiming Li, Baoyuan Wu, Longkang Li, Ran He, and Siwei Lyu. Invisible backdoor attack with sample-specific triggers. In ICCV, pp. 16463–16472, 2021d
> > > >
> > > >
> > > > [5] Kang Liu, Brendan Dolan-Gavitt, and Siddharth Garg. Fine-pruning: Defending against backdooring attacks on deep neural networks. In International Symposium on Research in Attacks, Intrusions, and Defenses, pp. 273–294. Springer, 2018
> > > >
> > > > [6] Yansong Gao, Change Xu, Derui Wang, Shiping Chen, Damith C Ranasinghe, and Surya Nepal. Strip: A defence against trojan attacks on deep neural networks. In Proceedings of the 35th Annual Computer Security Applications Conference, pp. 113–125, 2019.
> > > >
> > > > [7] Bolun Wang, Yuanshun Yao, Shawn Shan, Huiying Li, Bimal Viswanath, Haitao Zheng, and Ben Y Zhao. Neural cleanse: Identifying and mitigating backdoor attacks in neural networks. In 2019 IEEE Symposium on Security and Privacy (SP), pp. 707–723. IEEE, 2019.
> > > >
> > > > [8] Yige Li, Xixiang Lyu, Nodens Koren, Lingjuan Lyu, Bo Li, and Xingjun Ma. Anti-backdoor learning: Training clean models on poisoned data. Advances in Neural Information Processing Systems, 34, 2021a.
> > > >
> > > > [9] Yige Li, Xixiang Lyu, Nodens Koren, Lingjuan Lyu, Bo Li, and Xingjun Ma. Neural attention distillation: Erasing backdoor triggers from deep neural networks. arXiv preprint arXiv:2101.05930, 2021b.

---

> > > > > ### Comment · Reviewer_YTpa · 2022-11-24
> > > > > **Thanks for the responses.**
> > > > >
> > > > > 1) Thanks for trying to equalize the threat models across different attacks. I appreciate that you evaluated the k-trigger attack without adaptive regularization. I think it's necessary to evaluate TaCT with k-triggers as well since that's the closest prior work to yours.
> > > > >
> > > > > 2) "we do not consider the (correctly labeled) regularization samples malicious", I'm not convinced with this assumption. Any out-of-distribution sample the attacker crafts and manages to inject into the defenders data set is malicious and is poison. This is how these attacks are performed in practice. Thank you for still running the experiments, though.
> > > > >
> > > > > 3) If this is the case "more malicious payload samples empirically lead to less stealthiness", then you need to evaluate the other attacks with fewer samples not more samples. You have an experiment in Table 3a for vanilla blending attack, but what about TaCT with fever samples (with less than 150+150 samples). Is there a spot where TaCT becomes almost as undetectable as your attack? So, if you found that the sweet spot for your attack is 150+150 poisons to balance between ASR and undetectability, what would be the sweet spot for TaCT?
> > > > >
> > > > > 4) You can tune AC by increasing the #of clusters (the default is 2, but if you make that 10, then you'll eliminate fewer samples in general). Or by using DBSCAN, instead of k-means, you can directly give the min cluster size as a hyper-parameter, which would determine the #samples eliminated. I acknowledge that this could be a non-trivial change.

---

> > > > > > ### Author Response · Authors · 2022-11-27
> > > > > > **Author Response (Round II)**
> > > > > >
> > > > > > Thank you for providing further comments on our additional experiments!
> > > > > >
> > > > > > **Q1:**  Evaluating TaCT with k-triggers as well.
> > > > > >
> > > > > > **A1:** Thank you for suggesting TaCT with k-triggers. We have supplemented additional results on this case below.
> > > > > >
> > > > > > **Q2:** Any out-of-distribution sample the attacker crafts and manages to inject into the defenders data set is malicious and is poison.
> > > > > >
> > > > > > **A2:** We totally agree with you on this point. Thus, in our previous rebuttal, we have supplemented results where all attacks use the same number of poison samples (counting both payload and regularization samples). On the other hand, we are sorry for any misunderstandings. In our context, by saying "not malicious", we mean these regularization samples do not induce latent separation. Therefore, equilizing the number of payload samples would be the fairest way to compare latent separation. If we equilize the number of total poison samples (counting both payload / regularization), then non-adaptive attacks will have higer number of payload samples (stronger latent separation) and our adaptive attacks will have lower number of payload samples (weaker latent separation).
> > > > > >
> > > > > > **Q3:** Sweet spot for TaCT.
> > > > > >
> > > > > > **A3:** We appreciate your further concern about more experiments and comparisons with TaCT. Here we cast **TaCT-k**, combining TaCT with the same k triggers setting in our Adap-Patch. In addition, we explore whether fewer samples would help TaCT and TaCT-k evade latent-space detection. Our results are shown in the table below.
> > > > > >
> > > > > > | Attack $\downarrow$ | #Payload Samples $\rightarrow$ | 50 | 100 | 150 |
> > > > > > | --- | --- | --- | --- | --- |
> > > > > > | TaCT | ASR (CA)  w/o defense | 93.9 (91.8) | 96.9 (91.5) | 95.1 (91.7) |
> > > > > > | | SPECTRE ER (SA) | 100.0 (0.1) | 100.0 (0.1) | 100.0 (0.2) |
> > > > > > | TaCT-k | ASR (CA)  w/o defense | 95.4 (92.0) | 100.0 (91.9) | 100.0 (91.8) |
> > > > > > | | SPECTRE ER (SA) | 90.0 (0.1) | 95.0 (0.1) | 99.3 (0.2) |
> > > > > > | **Adap-Blend** | ASR (CA)  w/o defense | 18.1 (91.8) | 58.2 (91.4) | 76.5 (91.6) |
> > > > > > | | SPECTRE ER (SA) | 2.0 (0.1) | 0.0 (0.3) | 6.9 (0.5) |
> > > > > > | **Adap-Patch** | ASR (CA)  w/o defense | 75.1 (91.6) | 83.0 （91.8) | 97.5 (91.9) |
> > > > > > | | SPECTRE ER (SA) | 6.0 (0.1) | 0.0 (0.3) | 0.0 (0.5) |
> > > > > >
> > > > > > ("ASR" for attack success rate, "CA" for clean accuracy, "ER" for elimination rate, "SR" for sacrifice rate. Following our Adap-Blend and Adap-Patch, for TaCT, we use $\rho_\text{regularization} = \rho_\text{payload}$; for TaCT-k, we use $\rho_\text{regularization} = 2 * \rho_\text{payload}$. We only show defense results of SPECTRE since it consistently provides the highest poison elimination.)
> > > > > >
> > > > > > Clearly, even with fewer poison samples and the asymmetric k-triggers, TaCT (TacT-k) cannot evade latent-space detection ($\ge 90$\% payload samples are cleansed by SPECTRE). On the other hand, both our adaptive attacks can hardly be detected (only $<7$\% payload samples are cleansed by SPECTRE), and thus significantly outperform TaCT w.r.t. stealthiness against latent space defenses.
> > > > > >
> > > > > >
> > > > > > **Q4:** Improvements on clustering analysis.
> > > > > >
> > > > > > **A4:** We sincerely thank the reviewer for suggesting additional clustering analysis techniques to build better defenses with good calibration. We will definitely try to incorporate these suggestions into our final version. On the other hand, we want to explain that --- our paper is an attack paper, thus, initially we believe using the original implementations of previous defenses would be the most standard practice. Still, we do understand the reviewer's concerns. We would incorporate these suggestions.

---

> ### Author Response · Authors · 2022-11-20
> **Thanks to Reviewer YTpa**
>
> We would like to thank you again for reviewing our work and the valuable feedback, and in particular for recognizing the strengths of our paper in terms of formulation, novelty, valuable contributions, and paper writing.
>
> Please kindly let us know if you have any additional questions or require further clarification. We are happy to address them before the rebuttal ends.

---

> ### Comment · Reviewer_YTpa · 2022-11-29
> **Response to your response to my response**
>
> To avoid having too deep of a comment chain, I'm replying to my own comment.
>
> Thanks for your replies and more experiments, it looks pretty convincing now. I'll update my score and please update your main paper (not just the appendix) to discuss these new results. I think that would make your paper stronger.
>
> Best of luck.

---

> > ### Author Response · Authors · 2022-11-30
> > **Thank you for your positive feedback (and summary of our rebuttal)**
> >
> > Thank you again for your valuable time, comments, and suggestions on
> > (1) Fairness of experimental comparison,
> > (2) Calibration of baseline defenses,
> > (3) Ablation study on the necessity of regularization samples,
> > (4) Additional evaluations on other baseline defenses not based on latent separation
> > These comments and suggestions greatly help us to improve the quality of our paper. During the rebuttal period, we supplement additional results and discussions to address the fairness issues of our comparison. We also add additional ablation to further show the necessity of regularization samples. New results that evaluate against other defenses not based on latent separation also make our work more comprehensive. We are glad that our efforts have addressed your major concerns. Your recognition of our paper encourages us a lot! In our final version, we will wrap up our additional results in the main paper. Thanks again for your great suggestions!
> >
> > Best,
> > Paper3230 Authors

---

### Official Review · Reviewer_sY26 · 2022-10-23

**Confidence:** 4
**Correctness:** 3
**Technical Novelty And Significance:** 3
**Empirical Novelty And Significance:** 3
**Recommendation:** 6

**Clarity, Quality, Novelty And Reproducibility:**

The proposed attack is novel. The code is provided in the submission for reproducibility.

**Strength And Weaknesses:**

Strengths:
1. The proposed adaptive attack is well-motivated by the two insights and aims to avoid the latent separability assumption to circumvent the existing defenses.
2. The evaluation shows the superior attack performance of the proposed adaptive attacks, compared with the existing attacks against four state-of-the-art latent separation defenses.
3. The paper is well-written and easy to follow.

Weaknesses:
1. The scope of this work is somewhat limited. The paper only investigates the defenses built upon the latent separation assumption. Other effective defenses, e.g., trigger synthesis, are not examined in the paper.
2. The proposed attacks become ineffective when the poison ratio is high. It could be one limitation of the attacks when a relatively high poison ratio is required in the attack.
3. Since the attacks apply strong triggers in the test time and weakened triggers in the training data, will the asymmetric triggers become easier to detect in the test time?
4. It would be great to increase the font size in the figures (Figure 4 and 5).


**Summary Of The Paper:**

This paper investigates the latent separability assumption in backdoor defenses. The paper presents adaptive backdoor poisoning attacks against this assumption and circumvents the recent defenses.

**Summary Of The Review:**

The paper investigates a pervasive assumption in many backdoor defenses -  the defenses aim to identify the separation of latent features in backdoor attacks. The proposed adaptive backdoor poisoning attacks are well-motivated and effective in countering the defenses. However, the scope of the work is limited. The paper does not consider other defenses that are not based on the latent separation assumption.

---

> ### Author Response · Authors · 2022-11-14
> **Author Response (sY26) --- Part I**
>
> We sincerely thank you for your valuable time and comments. We are encouraged by your positive comments on our **motivation**, **valuable contribution**, and **paper writing**! We will alleviate your remaining concerns in this rebuttal, as follows.
>
> ---
>
> **Q1**: The scope of this work is somewhat limited. The paper only investigates the defenses built upon the latent separation assumption. Other effective defenses, e.g., trigger synthesis, are not examined in the paper.
>
> **Q3**: Since the attacks apply strong triggers in the test time and weakened triggers in the training data, will the asymmetric triggers become easier to detect in the test time?
>
> **A1 & A3**: Thank you for raising the concerns about not including results of other defenses and test-time stealthiness of our adaptive attacks! Since the topic of this paper is to revisit the latent separability assumption, we mainly put evaluations results against latent separation based defense. We appreciate the reviewer for suggesting additional evaluations against defenses that are not based on latent separation. To address the reviewers' concerns, hereby, we supplement **other defenses**, including:
> 1. Fine Pruning (FP[1]), Neural Cleanse (NC[3]), Neural Attention Distillation (NAD[5]) that aim to remove the backdoor. We report Attack Success Rate (ASR) and Clean Accuracy (CA) after these defenses.
> 2. STRIP\(C\) and STRIP(F), the two variants of STRIP [2] for detecting backdoor training samples and filtering test time backdoor samples respectively. We report the elimination rate (ER) of backdoor poison samples and sacrifice rate of clean samples (SR).
> 3. Anti-backdoor Learning (ABL[4]) that isolates backdoor training samples based on the speed of fitting. We report its isolation precision (IP) used by the original paper.
>
> | Defenses$\rightarrow$ | FP [1] | STRIP \(C\) [2] | STRIP (F) [2] | NC [3] | ABL [4] | NAD [5] |
> | -------- | -------- | -------- | -------- | -------- | -------- | -------- |
> | **Attacks $\downarrow$**     | **ASR (CA)** | **ER (SR)**  | **ER (SR)**  | **ASR (CA)** | **IP**  | **ASR (CA)** |
> | Blend | 78.1 (81.1) | 17.3 (9.7) | 14.3 (10.0) | 87.5 (91.9) | 0.4 | 4.9 (80.8) |
> | **Adap-Blend** | 77.5 (76.9) | 0.7 (9.7) | 7.4 (10.0) | 72.4 (91.5) | 0.0 | 9.5 (81.2) |
> | BadNet | 88.9 (80.7) | 100.0 (10.1) | 100.0 (10.0) | 1.7 (90.3) | 4.2 | 3.1 (83.1) |
> | **Adap-Patch** | 99.5 (80.9) | 21.3 (10.6) | 99.9 (10.0) | 2.2 (89.2) | 0.0 | 19.5 (81.3) |
>
> ("ASR" for attack success rate, "CA" for clean accuracy, "ER" for elimination rate, "SR" for sacrifice rate, "IP" for isolation precision of poison samples, "STRIP \(C\) \& (F)" for STRIP as a poison cleanser and an input filter. Refer to our Appendix D for more details.)
>
> As shown, compared to the naive versions of attacks (Blend \& BadNet), the attacks with our adaptive strategies (Adap-Blend \& Adap-Patch) still consistently show stronger or at least comparable resistance to those non-latent-separation-based defenses we evaluate against. For example, against STRIP \(C\), our adaptive attacks even significantly outperform the corresponding naive attacks -- possibly due to the weaker correlation between the backdoor trigger and the target label in our poison samples.
>
> Above results also **address the reviewer's concern on test-time defenses**:
> 1. Against STRIP (F), a **test-time** backdoor input filter, though our asymmetric triggers would enhance the backdoor signal, the poison samples are not more susceptible to be detected at test time (compared to the naive attacks without our adaptive strategy). STRIP (F)[2] detects 14.3\% and 100.0\% poison samples of Blend \& BadNet, and detects 7.4\% and 99.9\% poison samples of Adap-Blend \& Adap-Patch. This implies: 1) our adaptive strategy does not decrease test-time stealthiness; 2) with wisely chosen triggers (e.g. blending HelloKitty), the test-time backdoor detection could further be evaded.
> 2. Besides, if aware of the test-time backdoor detection, the Adap-Patch adversary can choose some other weaker test-time triggers (see the table below and Appendix D) to achieve tradeoff between ASR and test-time stealthiness (against STRIP (F)).
>
>     | Triggers | ASR | Elimination | Sacrifice |
>     | -------- | -------- | -------- | -------- |
>     | 1st Trigger + 3nd Trigger | 97.5 | 99.0 | 10.0 |
>     | 2nd Trigger + 4th Trigger | 86.5 | 70.0 | 10.0 |
>     | 1st Trigger (opacity=0.5) + 2nd Trigger (opacity=0.7) | 59.7 | 35.0 | 10.0 |
>     | 2nd Trigger | 25.6 | 5.3 | 10.0 |

---

> > ### Author Response · Authors · 2022-11-14
> > **Author Response (sY26) --- Part II**
> >
> > ---
> >
> > **Q2**: The proposed attacks become ineffective when the poison ratio is high. It could be one limitation of the attacks when a relatively high poison ratio is required in the attack.
> >
> > **A2**: The reviewer mentions our attack could be more susceptible to latent-space defenses when the poison ratio is high. We agree with the reviewer that our attacks' stealthiness in the latent representation space would degrade when the poison rate is high, which could be a limitation.
> > 1. However, this is the limitation of most (if not all) backdoor poisoning attacks --- a high poison rate means a large amount of training samples would contain a rigid trigger pattern. This trigger pattern would thus definitely be of high statistical significance. Then, empirically, models tend to learn a strong signal in the feature space for this statistically significant pattern. Besides, with increasing statistical significance, other statistical analyses (even human inspection) can also identify poison samples more easily.
> > 2.  On the other hand, as shown in both Table 1 and Figure 1 of our paper --- even with a low poison rate, other existing backdoor attacks still exhibit strong latent separation. In contrast, with the same number of payload poison samples, our attacks exhibit the best stealthiness in the latent representation space.
> > 3. Besides, though with a low poison rate, our attacks still achieve high ASR. In practice, if our attacks with this low poison rate are already successful, attackers usually have no incentive to increase the poison rate.
> >
> >
> > ---
> >
> > **Q4**: It would be great to increase the font size in the figures (Figure 4 and 5).
> >
> > **A4**: Thank you for pointing it this flaw! We have increased the font size. In addition, we have carefully gone through the whole paper and polished it again.
> >
> > [1] Kang Liu, Brendan Dolan-Gavitt, and Siddharth Garg. Fine-pruning: Defending against backdooring attacks on deep neural networks. In International Symposium on Research in Attacks, Intrusions, and Defenses, pp. 273–294. Springer, 2018
> >
> > [2] Yansong Gao, Change Xu, Derui Wang, Shiping Chen, Damith C Ranasinghe, and Surya Nepal. Strip: A defence against trojan attacks on deep neural networks. In Proceedings of the 35th Annual Computer Security Applications Conference, pp. 113–125, 2019.
> >
> > [3] Bolun Wang, Yuanshun Yao, Shawn Shan, Huiying Li, Bimal Viswanath, Haitao Zheng, and Ben Y Zhao. Neural cleanse: Identifying and mitigating backdoor attacks in neural networks. In 2019 IEEE Symposium on Security and Privacy (SP), pp. 707–723. IEEE, 2019.
> >
> > [4] Yige Li, Xixiang Lyu, Nodens Koren, Lingjuan Lyu, Bo Li, and Xingjun Ma. Anti-backdoor learning: Training clean models on poisoned data. Advances in Neural Information Processing Systems, 34, 2021a.
> >
> > [5] Yige Li, Xixiang Lyu, Nodens Koren, Lingjuan Lyu, Bo Li, and Xingjun Ma. Neural attention distillation: Erasing backdoor triggers from deep neural networks. arXiv preprint arXiv:2101.05930, 2021b.

---

> ### Author Response · Authors · 2022-11-20
> **Thanks to Reviewer sY26**
>
> We would like to thank you again for reviewing our work and the valuable feedback, and in particular for recognizing the strengths of our paper in terms of motivation, valuable contribution, and paper writing.
>
> Please kindly let us know if you have any additional questions or require further clarification. We are happy to address them before the rebuttal ends.

---

> ### Author Response · Authors · 2022-11-30
> **Thanks to Reviewer sY26**
>
> We would like to thank you again for reviewing our work and the valuable feedback, and in particular for recognizing the strengths of our paper in terms of motivation, valuable contribution, and paper writing.
>
> During the rebuttal period, we:
>
> 1. Added additional evaluation results against other defenses that are not based on latent separation, which makes our work more comprehensive. In particular, we evaluated against test time defense as you suggested. We show that 1) our adaptive strategy does not decrease test-time stealthiness; 2) with wisely chosen triggers (e.g. blending HelloKitty), the test-time backdoor detection could further be evaded.
>
> 2. Discussed the issue of our reliance on a low poison rate and justified why it is reasonable.
>
> 3. Fixed typos and some flaws in our presentation.
>
> We hope these efforts can further address your concerns. Please kindly let us know if you have any additional questions or require further clarification. We are happy to address them before the rebuttal ends.

---

> ### Author Response · Authors · 2022-12-05
> **A Gentle Reminder of the Final Feedback**
>
> We would like to thank the reviewer for the helpful discussion during the first round of the review. We hope our response has adequately addressed your previous comments and concerns. We take this as a great opportunity to improve our work and shall be grateful for any additional feedback you could give to us.

---

### Official Review · Reviewer_4n1q · 2022-10-25

**Confidence:** 3
**Correctness:** 4
**Technical Novelty And Significance:** 3
**Empirical Novelty And Significance:** 3
**Recommendation:** 6

**Clarity, Quality, Novelty And Reproducibility:**

The work is clear and of high quality. I am not an expert on backdoor attacks and thus I cannot judge the novelty and originality (the only similar attack I have seen is the m-way attack cited above, but it's possible there are more attacks aimed at circumventing latent separability that I'm not aware of).

**Strength And Weaknesses:**

Strengths:

- The approach is well-motivated and the results are convincing and thorough. The paper evaluates against state-of-the-art latent separability attacks and succeeds against all of them.
- The writing and motivation is clear.
- The authors conduct ablation studies to pinpoint the sources of indistinguishability.

Weaknesses:

- I am a bit unclear on the difference between the Adaptive-Blend attack and the m-way attack from Xie et al (https://openreview.net/forum?id=rkgyS0VFvr)
- Figure 4 suggests that the backdoor attack may not be truly "indistinguishable" in the sense that knowing which samples are poisoned in advance allows one to train a somewhat reliable classifier for finding the backdoor samples. Is there any formal sense in which the samples are indistinguishable (beyond just circumventing latent separability-based attacks?)
- A relevant baseline might be to use a method like Tan et al (I think cited in this paper as Shokri et al) https://arxiv.org/abs/1905.13409 but using a separate training algorithm to the one used by the defender (this would be something like a "substitute model attack" in adversarial robustness).


**Summary Of The Paper:**

Motivated by the observation that backdoor attacks tend to cluster in the latent space of neural networks, this paper proposes a set of adaptive attacks whose poison samples are indistinguishable (in terms of representation) from clean samples.

**Summary Of The Review:**

This paper proposes a set of backdoor attacks that circumvent the latent separability assumption. The corresponding backdoor attack circumvent a variety of latent separability-based assumptions.

---

> ### Author Response · Authors · 2022-11-14
> **Author Response (4n1q) --- Part I**
>
> We sincerely thank you for your valuable time and comments. We are encouraged by your positive comments on our **motivation**, **comprehensive and convincing results**, **useful ablation studies**, and **paper writing**! We will alleviate your remaining concerns in this rebuttal, as follows.
>
>
> ---
> **Q1**: I am a bit unclear on the difference between the Adaptive-Blend attack and the m-way attack from Xie et al (https://openreview.net/forum?id=rkgyS0VFvr)
>
> **A1**: We thank the reviewer for mentioning the m-way attack by Xie et al. [1]. They proposed DBA targeting federated learning scenarios, and also studied a type of asymmetric triggers for data poisoning and testing time attacks respectively. Compared with our work, (1) m-way attack still suffers from latent separation; (2) it didn't consider regularization in data poisoning as well as its implication for suppressing latent separation characteristics.
>
> To highlight the differences, we:
> 1. Supplement an additional evaluation on the m-way attack on cifar10. We follow the original settings of Xie et al., and use a same number of payload samples to that of our main evaluation in Table 1.
> 2. Further implement an adaptive version of the m-way attack (Adap-M-Way) via incorporating our idea of regularization samples. Number of payload and regularization samples are also the same to that of Adaptive-Blend in Table 1.
>
> As shown below, the original m-way attack proposed by Xie et al. can be defended (ASR < 5%) by 3 out of 4 latent separation based defenses that we evaluate against. After being adapted with our regularization techniques, none of the four attacks can defend it.
>
> Table: Results of M-way Attack \& Adap-M-Way Attack
> |          |          | M-way    | Adap-M-way |
> | -------- | -------- | -------- | -------- |
> | None | ASR | 87.8 | 73.8 |
> | | CA     | 91.3 | 91.0 |
> | SS          | Elimination | 33.3 | 19.0 |
> |           | Sacrifice | 4.4 | 4.5|
> |           | ASR | 4.7 | **66.6** |
> |           | CA | 91.3 | 90.9 |
> | AC          | Elimination | 0.0 | 0.0 |
> |           | Sacrifice | 0.0 | 0.0 |
> |           | ASR | **87.8** | **73.8** |
> |           | CA | 91.3 | 91.0 |
> | SCAn          | Elimination | 55.3 | 0.0 |
> |           | Sacrifice | 7.5 | 0.0 |
> |           | ASR | 3.3 | **73.8** |
> |           | CA | 91.0 | 91.0 |
> | SPECTRE          | Elimination | 64.0 | 10.0 |
> |           | Sacrifice | 0.3 | 0.4 |
> |           | ASR | 2.4 | **62.5** |
> |           | CA | 91.4 | 91.5 |
>
> ("ASR" for attack success rate, "CA" for clean accuracy. We follow the same poison rate settings in our main experiment.)
>
> ---
>
> **Q2**: Figure 4 suggests that the backdoor attack may not be truly "indistinguishable" in the sense that knowing which samples are poisoned in advance allows one to train a somewhat reliable classifier for finding the backdoor samples. Is there any formal sense in which the samples are indistinguishable (beyond just circumventing latent separability-based attacks?)
>
> **A2**: We thank the reviewer for suggesting the formal sense of "indistinguishability". Besides circumventing latent separability-based attacks, we do provide some other senses of indistinguishability in our paper.
>
> 1. Qualitatively, Fig.1 and Fig.4 illustrate the extent of separation in a low dimension projection plane.
> 2. Quantitatively, in Table 1, we report the **Silhouette Score (Coefficient)** [2,3] that formally measures the distance between poison and clean clusters in our context. As shown below, our Adap-attacks have significantly lower silhouette scores, which means the "distance" between clean and poison clusters are closer (and are thus more indistinguishable).
>
>     Table: Silhouette Scores of Attacks.
>     | Blend | BadNet | ISSBA | Dynamic | CL | TaCT | **Adap-Blend** | **Adap-Patch** |
>     | -------- | -------- | -------- | -------- | -------- | -------- | -------- | -------- |
>     | 0.2608 | 0.4744 | 0.3933 | 0.4358 | 0.3964 | 0.2866 | **0.1065** | **0.0856** |

---

> > ### Author Response · Authors · 2022-11-14
> > **Author Response (4n1q) --- Part II**
> >
> > **Q3**: A relevant baseline might be to use a method like Tan et al (I think cited in this paper as Shokri et al) https://arxiv.org/abs/1905.13409 but using a separate training algorithm to the one used by the defender (this would be something like a "substitute model attack" in adversarial robustness).
> >
> > **A3**: We thank the reviewer for suggesting Tan et al. [4] as a baseline.
> > 1. As we have discussed in Section 2 of our paper, Tan et al. work on a different threat model than ours. We focus on poison-only backdoor attacks where adversaries only control a small portion of training data. However, Tan et al. assume a stronger threat model where adversaries can also control the training process of victim models --- so they can directly encode the latent indistinguishability as an objective into the training loss.
> > 2. On the other hand, we agree that it would be indeed helpful to measure the "latent indistinguishability" (e.g. report the Silhouette Score) of Tan et al. as well. In some sense, such attacks with a stronger threat model can suppress the latent separation more easily. The "latent indistinguishability" achieved by such stronger backdoor attacks can thus give us an upper bound on the "indistinguishability" that might be achieved by the weaker poison-only backdoor attacks. We will add this comparison in our final version
> >
> > [1] Xie, Chulin, et al. "Dba: Distributed backdoor attacks against federated learning." International Conference on Learning Representations. 2019.
> >
> > [2] Peter J Rousseeuw. Silhouettes: a graphical aid to the interpretation and validation of cluster analysis. Journal of computational and applied mathematics, 20:53–65, 1987.
> >
> > [3] https://scikit-learn.org/stable/modules/generated/sklearn.metrics.silhouette_score.html
> >
> > [4] Shokri, Reza. "Bypassing backdoor detection algorithms in deep learning." 2020 IEEE European Symposium on Security and Privacy (EuroS&P). IEEE, 2020.

---

> ### Author Response · Authors · 2022-11-20
> **Thanks to Reviewer 4n1q**
>
> We would like to thank you again for reviewing our work and the valuable feedback, and in particular for recognizing the strengths of our paper in terms of motivation, comprehensive and convincing results, useful ablation studies, and paper writing.
>
> Please kindly let us know if you have any additional questions or require further clarification. We are happy to address them before the rebuttal ends.

---

> ### Author Response · Authors · 2022-11-30
> **Thanks to Reviewer 4n1q**
>
> We would like to thank you again for reviewing our work and the valuable feedback, and in particular for recognizing the strengths of our paper in terms of motivation, comprehensive and convincing results, useful ablation studies, and paper writing.
>
> During the rebuttal period, we:
>
> (1) Added additional discussions on the M-way attack that you mentioned, and added additional experimental comparison. We hope this could further clarify the novelty of our work.
>
> (2) Supplemented discussion and results on the formal sense of latent indistinguishability.
>
> (3) Discussed the different threat model of Tran et al. compared with ours. We also agreed that it would be helpful to measure the "latent indistinguishability" (e.g. report the Silhouette Score) of Tan et al. as well. The "latent indistinguishability" achieved by such stronger backdoor attacks can hopefully give us an upper bound on the "indistinguishability" that might be achieved by the weaker poison-only backdoor attacks. We will add this comparison in our final version.
>
> We hope these efforts can further address your concerns. Please kindly let us know if you have any additional questions or require further clarification. We are happy to address them before the rebuttal ends.

---

> ### Author Response · Authors · 2022-12-05
> **A Gentle Reminder of the Final Feedback**
>
> We would like to thank the reviewer for the helpful discussion during the first round of the review. We hope our response has adequately addressed your previous comments and concerns. We take this as a great opportunity to improve our work and shall be grateful for any additional feedback you could give to us.

---

### Official Review · Reviewer_XjVu · 2022-11-05

**Confidence:** 5
**Correctness:** 3
**Technical Novelty And Significance:** 2
**Empirical Novelty And Significance:** 3
**Recommendation:** 3

**Clarity, Quality, Novelty And Reproducibility:**

The paper is well written. The the work can be reproduced given the description and the code.

**Strength And Weaknesses:**

The motivation of the paper is well-described. The paper extends the work in some backdoor attack methods to data poisoning, which is a different threat model. This could be a valuable contribution to the backdoor research community. However, I think that there are some major issues in the paper, especially in its similarity to an existing work in the backdoor domain:

- One of the 2 main parts (regularization samples) of the proposed method exhibits a close similarity to Peng et al's work. This idea has been already explored and studied extensively before. Peng et al. also reduce the strength of the association between the trigger and the target using the exact same method, but with a more extensive theoretical analysis.
- While the proposed method is effective, I think that the experiments also show that other existing backdoor attacks are also pretty effective against most latent-space defenses. However, the paper does not show the results of the existing methods on other studied datasets, besides only CIFAR10.
- The paper should include evaluations against other defense methods. This is a new type of attack, thus it should be validated not only against latent-space defenses but also other types of defenses, or at least include any insight on whether the method can bypass other defenses. Making it pass latent-space defenses may make it more detectable in other defenses.

some minor grammatical errors: "while keep" (threat model)

Peng et al. Label-Smoothed Backdoor Attack. https://arxiv.org/abs/2202.11203

**Summary Of The Paper:**

The paper investigates the latent separability of the existing backdoor attacks, which makes them easily caught by existing defenses that look at the latent space. The paper then crafts an attack using data poisoning without latent separability. The proposed adaptive backdoor attack method uses regularized samples (samples with triggers but correct labels) and asymmetric triggers that have different intensities in training and inference. The proposed attack is shown to escape existing latent-space defenses on CIFAR10, GTSRB, and 10-class subset of Imagenet.

**Summary Of The Review:**

The paper has made some important contributions but its similarity to some existing work makes it less novel. Therefore, it is not strong enough for an ICLR acceptance recommendation.

---

> ### Author Response · Authors · 2022-11-14
> **Author Response (XjVu) --- Part I**
>
> We sincerely thank you for your valuable time and comments. We are encouraged by your positive comments on our **motivation**, **practical threat model**, **valuable contribution**, and **paper writing**! We will alleviate your remaining concerns in this rebuttal, as follows.
>
>
> ---
> **Q1**: Similarities to (Peng et al., 2022) [1]
>
> **A1**: We sincerely thank you for bringing this arXiv paper to our notice. After a careful review on (Peng et al.,2022) [1], we agree that it bears similarity with some aspects of our work. However, we argue that the mere presence of this paper on Arxiv should not be a basis for rejecting our paper. The main reasons are as follows:
> - **The suggested paper is a concurrent work to this submission without being peer-reviewed**. According to the [reviewer guide](https://iclr.cc/Conferences/2023/ReviewerGuide) of ICLR 2023, authors can be excused for not knowing about papers not published in peer-reviewed conference proceedings or journals, which includes papers exclusively available on arXiv. This paper appeared online (on arXiv) this year, which is after the date we started working on this project, and has not been referred to by any of the existing literature.
> - **Our method has many fundamental differences in technical aspects, compared to (Peng et al., 2022)**.
>   1. **Motivation**. Our method is motivated by our observations that the latent separability is so pervasive across both classical and advanced backdoor poisoning attacks. (Peng et al., 2022) was inspired by the overfitting problem of existing backdoor attacks. However, they did not provide any definition or detailed examples of the overfitting problem. According to their experiments and method, we speculate that it means DNNs tend to predict poisoned samples as the target label with very high confidence (i.e., nearly 1), which is different from our motivation.
>   2. **The Generation of Regularization/Poisoned Samples**. In general, both our method and (Peng et al., 2022) preserve the ground-truth label (instead of re-assigning to the target label) of a fraction of modified samples. However, the fraction of our method is sample-independent (see our Eq.(2)) whereas that of (Peng et al., 2022) is sample-specific (see its Eq.(2)). This difference is because we have different mechanisms: we intend to alleviate the latent separability whereas they aimed to assign a soft-label to each poisoned sample.
>   3. **The Improvement Module**. In general, both our method and (Peng et al., 2022) have an additional module to further improve the main attack. However, we adopt asymmetric triggers to alleviate latent separability and encourage trigger diversity while preserving high attack effectiveness (see our Section 5.1), whereas (Peng et al., 2022) implanted multiple different backdoors to improve attack effectiveness (see its Section 3.3).
>   4. **Baseline Defenses**. We mainly evaluate our attacks against latent separation based defenses, because our goal is to design adaptive backdoor poisoning attacks against this family of defenses. In contrast, (Peng et al., 2022) only adopted STRIP [2] and Neural Cleanse [3] for evaluation. Neither of these two defenses is based on latent separation, and they are known to be ineffective against many existing advanced attacks (e.g., ISSBA [4]).
> - **Our method has unique contributions, compared to (Peng et al., 2022)**.
>   1. We demonstrate that the latent separability assumption holds across a diverse set of classical and advanced backdoor poisoning attacks in the existing literature. Our work thus highlights the big blank in designing backdoor poisoning attacks that are stealthy in the latent representation space, and might consequently motivate future research in this direction.
>   2. We formulate and reveal the latent separability assumption that is underlying many state-of-the-art defenses --- as we have said, this assumption is indeed very persistent across a diverse set of existing attacks.
>   3. In this work, we have done a proof-of-concept study to show that the latent separability assumption could fail. This conclusion suggests defense designers take caution when leveraging latent separation as an assumption in their defenses. We believe this can influence the design of backdoor defenses in the future.
> - **The correctness of theorems in (Peng et al., 2022) is questionable and their code repository is still empty.** We are unable to verify the correctness of its main theorem (we believe it is incorrect as stated). Besides, it refers readers to its repository for implementation details, but the repository is still empty... Having said that, we would be happy to further discuss this paper in detail.
>
> However, we do understand your concerns and respect the contribution of your suggested paper. We have added a detailed comparison between our paper and (Peng et al., 2022) and a concurrency statement in the appendix of our revision.

---

> > ### Author Response · Authors · 2022-11-14
> > **Author Response (XjVu) --- Part II**
> >
> > **Q2**: While the proposed method is effective, I think that the experiments also show that other existing backdoor attacks are also pretty effective against most latent-space defenses. However, the paper does not show the results of the existing methods on other studied datasets, besides only CIFAR10.
> >
> > **A2**: Thank you for this question and we do understand your concerns that other existing backdoor attacks may be resistant to some latent-space defenses. We hope the following discussions can address your concerns:
> > - We agree that some of the existing attacks are also resistant to some early latent-space defenses ($e.g.$, Spectral Signature [5] and Activation Clustering [6]) — the main reason is that the clustering analyses in these defenses are weak. However, **this does not mean these attacks do not suffer from latent separation**. Qualitatively, as have been visualized in Figure 1 and Figure 7 in our paper, these attacks exhibit clear latent separations. Quantitatively, these attacks also have high Silhouette Scores, as have been reported in Table 1 of our paper.
> > - On the other hand, the reason why we mainly evaluate other attacks on CIFAR10 is that --- many existing attacks and defenses are mainly implemented and optimized for CIFAR10, which is the most standard playground for backdoor studies. For example, CL[7] is only implemented for CIFAR, ISSBA[4] is not effective on GTSRB, etc. Besides, although SPECTRE is optimized to be nearly perfect on CIFAR10, it fails  on GTSRB... Still, to further address the reviewer's concerns, we also supplement additional results of other attacks (CL and ISSBA are omitted for above reasons) on GTSRB as follow:
> >
> > Table: Additional results on the GTSRB dataset.
> >
> > | Defenses$\rightarrow$ | No Defense | SS | AC | SCAn | SPECTRE |
> > | -------- | -------- | -------- | -------- | -------- | -------- |
> > | **Attacks $\downarrow$**     | **ASR (CA)** | **ER (SR)**  | **ER (SR)**  | **ER (SR)**  | **ER (SR)** |
> > | Dynamic | 100.0 (98.0) | 96.2 (17.8) | 75.9 (3.4) | 84.84 (3.2) | 0.0 (0.1) |
> > | TaCT | 100.0 (97.7) | 97.5 (17.8) | 0.0 (3.0) | 0.0 (0.0) | 0.0 (0.1) |
> > | Blend | 86.4 (97.6) | 70.9 (17.8) | 84.8 (6.9) | 0.0 (3.6) | 0.0 (0.1) |
> > | **Adap-Blend** | 82.2 (97.8) | 43.0 (17.9) | 0.0 (3.5) | 0.0 (2.3) | 0.0 (0.1) |
> > | BadNet | 99.5 (97.8) | 100.0 (25.2) | 97.7 (2.3) | 99.2 (0.5) | 100.0 (0.3) |
> > | **Adap-Patch** | 63.1 (97.8) | 74.4 (25.4) | 0.0 (0.3) | 35.6 (4.6) | 0.0 (0.1) |
> >
> > ("ASR" for attack success rate, "CA" for clean accuracy, "ER" for elimination rate, and "SR" for sacrifice rate.)
> >
> > As shown, on GTSRB, our adaptive attacks still consistently show stronger resistance to latent space defenses than other attacks.
> >
> > ---
> >
> > **Q3**: The paper should include evaluations against other defense methods. This is a new type of attack, thus it should be validated not only against latent-space defenses but also other types of defenses, or at least include any insight on whether the method can bypass other defenses. Making it pass latent-space defenses may make it more detectable in other defenses.
> >
> > **A3**: Thank you for this insightful comment regarding other defense methods! We only evaluated our method under latent-space defenses since our main target is to circumvent the assumption of latent separability that was regarded as fundamental and essential for poison-only backdoor attacks in previous works. However, we do understand your concern that our method may make it more detectable in other defenses. To address the reviewers' concerns, hereby, we supplement other defenses, including:
> > 1. Fine Pruning (FP[8]), Neural Cleanse (NC[10]), Neural Attention Distillation (NAD[12]) that aim to remove the backdoor. We report Attack Sucess Rate (ASR) and Clean Accuracy (CA) after these defenses.
> > 2. STRIP\(C\) and STRIP(F), the two variants of STRIP [9] for detecting backdoor training samples and filtering test time backdoor samples respectively. We report the elimination rate (ER) of backdoor poison samples and sacrifice rate of clean samples (SR).
> > 3. Anti-backdoor Learning (ABL[11]) that isolates backdoor training samples based on the speed of fitting. We report its isolation precision (IP) used by the original paper.

---

> > > ### Author Response · Authors · 2022-11-14
> > > **Author Response (XjVu) --- Part III**
> > >
> > > Table: The resistance to additional defenses other than latent-space-based ones.
> > > | Defenses$\rightarrow$ | FP [8] | STRIP \(C\) [9] | STRIP (F) [9] | NC [10] | ABL [11] | NAD [12] |
> > > | -------- | -------- | -------- | -------- | -------- | -------- | -------- |
> > > | **Attacks $\downarrow$**     | **ASR (CA)** | **ER (SR)**  | **ER (SR)**  | **ASR (CA)** | **IP**  | **ASR (CA)** |
> > > | Blend | 78.1 (81.1) | 17.3 (9.7) | 14.3 (10.0) | 87.5 (91.9) | 0.4 | 4.9 (80.8) |
> > > | **Adap-Blend** | 77.5 (76.9) | 0.7 (9.7) | 7.4 (10.0) | 72.4 (91.5) | 0.0 | 9.5 (81.2) |
> > > | BadNet | 88.9 (80.7) | 100.0 (10.1) | 100.0 (10.0) | 1.7 (90.3) | 4.2 | 3.1 (83.1) |
> > > | **Adap-Patch** | 99.5 (80.9) | 21.3 (10.6) | 99.9 (10.0) | 2.2 (89.2) | 0.0 | 19.5 (81.3) |
> > >
> > > ("ASR" for attack success rate, "CA" for clean accuracy, "ER" for elimination rate, "SR" for sacrifice rate, "IP" for isolation precision of poison samples, "STRIP \(C\) \& (F)" for STRIP as a poison cleanser and an input filter, respective.)
> > >
> > > As shown, compared to the naive versions of attacks (Blend \& BadNet), the attacks with our adaptive strategies (Adap-Blend \& Adap-Patch) still consistently show stronger or at least comparable resistance to those non-latent-separation-based defenses we evaluate against. For example, against STRIP \(C\), our adaptive attacks even significantly outperform the corresponding naive attacks -- possibly due to the weaker correlation between the backdoor trigger and the target label in our poison samples. We also show in our Appendix D that by selecting different test-time triggers, an Adap-Patch attacker can further evade the test-time STRIP (F) backdoor filter.
> > >
> > > ---
> > > **Q4**: some minor grammatical errors: "while keep" (threat model).
> > >
> > > **A4**: Thank you for pointing it out! We have corrected this typo in our revision. In addition, we have carefully gone through the whole paper and polished it again.
> > >
> > > ---
> > >
> > >
> > > [1] Peng, Minlong, et al. "Label-Smoothed Backdoor Attack." arXiv preprint arXiv:2202.11203 (2022).
> > >
> > > [2] Yansong Gao, Change Xu, Derui Wang, Shiping Chen, Damith C Ranasinghe, and Surya Nepal. Strip: A defense against trojan attacks on deep neural networks. In Proceedings of the 35th Annual Computer Security Applications Conference, pp. 113–125, 2019.
> > >
> > > [3] Bolun Wang, Yuanshun Yao, Shawn Shan, Huiying Li, Bimal Viswanath, Haitao Zheng, and Ben Y Zhao. Neural cleanse: Identifying and mitigating backdoor attacks in neural networks. In 2019 IEEE Symposium on Security and Privacy (SP), pp. 707–723. IEEE, 2019.
> > >
> > > [4] Li, Yuezun, et al. “Invisible backdoor attack with sample-specific triggers.” Proceedings of the IEEE/CVF International Conference on Computer Vision. 2021.
> > >
> > > [5] Tran, Brandon, Jerry Li, and Aleksander Madry. "Spectral signatures in backdoor attacks." Advances in neural information processing systems 31 (2018).
> > >
> > > [6] Chen, Bryant, et al. "Detecting backdoor attacks on deep neural networks by activation clustering." arXiv preprint arXiv:1811.03728 (2018).
> > >
> > > [7] Turner, Alexander, Dimitris Tsipras, and Aleksander Madry. "Label-consistent backdoor attacks." arXiv preprint arXiv:1912.02771 (2019).
> > >
> > > [8] Kang Liu, Brendan Dolan-Gavitt, and Siddharth Garg. Fine-pruning: Defending against backdooring attacks on deep neural networks. In International Symposium on Research in Attacks, Intrusions, and Defenses, pp. 273–294. Springer, 2018
> > >
> > > [9] Yansong Gao, Change Xu, Derui Wang, Shiping Chen, Damith C Ranasinghe, and Surya Nepal. Strip: A defense against trojan attacks on deep neural networks. In Proceedings of the 35th Annual Computer Security Applications Conference, pp. 113–125, 2019.
> > >
> > > [10] Bolun Wang, Yuanshun Yao, Shawn Shan, Huiying Li, Bimal Viswanath, Haitao Zheng, and Ben Y Zhao. Neural cleanse: Identifying and mitigating backdoor attacks in neural networks. In 2019 IEEE Symposium on Security and Privacy (SP), pp. 707–723. IEEE, 2019.
> > >
> > > [11] Yige Li, Xixiang Lyu, Nodens Koren, Lingjuan Lyu, Bo Li, and Xingjun Ma. Anti-backdoor learning: Training clean models on poisoned data. Advances in Neural Information Processing Systems, 34, 2021a.
> > >
> > > [12] Yige Li, Xixiang Lyu, Nodens Koren, Lingjuan Lyu, Bo Li, and Xingjun Ma. Neural attention distillation: Erasing backdoor triggers from deep neural networks. arXiv preprint arXiv:2101.05930, 2021b.

---

> > > > ### Comment · Reviewer_XjVu · 2022-11-20
> > > > **Comments on additional results**
> > > >
> > > > Thank you for providing the additional results.
> > > >
> > > > * The study of backdoor attacks is to ensure that the deployment of ML models can be less vulnerable to these attacks. As mentioned by the authors in their response, previous works that only studied CIFAR10, poorly generalize to more realistic datasets (even doesn't work on GTSRB). In recent years, many backdoor papers evaluate their works on realistic datasets (including ImageNet/TinyImageNet), which becomes the norm for any study in the backdoor domain.
> > > >
> > > > * As I can observe from the new results (e.g. ASR) on GTSRB, the ASRs are falling more, which shows that the attack is less effective compared to the study on CIFAR10. Furthermore, compared to some methods such as TaCT, the proposed method is much less effective w.r.t. latent space defenses on GTSRB, further asking the question that whether latent space separation is actually an issue for realistic datasets and the proposed method could alleviate the robustness of the attacks against these defenses.

---

> > > > > ### Author Response · Authors · 2022-11-20
> > > > > **Author Response (Round-II)**
> > > > >
> > > > > Thank you for providing further comments on our additional experiments!
> > > > >
> > > > > ---
> > > > > **Q1**: The study of backdoor attacks is to ensure that the deployment of ML models can be less vulnerable to these attacks. As mentioned by the authors in their response, previous works that only studied CIFAR10, poorly generalize to more realistic datasets (even doesn't work on GTSRB). In recent years, many backdoor papers evaluate their works on realistic datasets, which becomes the norm for any study in the backdoor domain.
> > > > >
> > > > > **R1**: Thank you for supporting our opinion! We totally agree with you that studying backdoor attacks on different datasets has and should become the norm for new studies in the backdoor domain. Thus, in our paper, we have evaluated our attacks on three different datasets (CIFAR10, GTSRB, Imagenette) for verifying their effectiveness.
> > > > >
> > > > > ---
> > > > > **Q2**: As I can observe from the new results (e.g. ASR) on GTSRB, the ASRs are falling more, which shows that the attack is less effective compared to the study on CIFAR10.
> > > > >
> > > > >
> > > > > **R2**: Thank you for your comments and we do understand your concerns. We believe that these differences may not hinder our observations and contributions. We would like to alleviate your concerns as follows:
> > > > >
> > > > > - Adap-Patch achieves lower ASR on GTSRB than that on CIFAR10 --- but the ASR is still over 60%, which is considerably high. For example, in autonomous driving, even a 10% chance (which is $\ll 60\%$) of attacking a "no passing" to a "60 km/h speed limit" poses a significant safety threat.
> > > > > - Compared with BadNet, the non-adaptive patch-based attack, Adap-Patch also exhibits stronger resistance to all latent-separation-based defenses.
> > > > > - Adap-Blend (another instance of our adaptive attacks) achieves an even higher ASR (>80%) comparable to that of its non-adaptive counterpart (Blend), and Adap-Blend exhibits the strongest resistance to all latent-separation-based defenses among all evaluated attacks.
> > > > >
> > > > >
> > > > > ---
> > > > >
> > > > > **Q3**: Furthermore, compared to some methods such as TaCT, the proposed method is much less effective w.r.t. latent space defenses on GTSRB, further asking the question that whether latent space separation is actually an issue for realistic datasets and the proposed method could alleviate the robustness of the attacks against these defenses.
> > > > >
> > > > >
> > > > > **R3**: Thank you for highlighting the results of TaCT on GTSRB. We do understand your concerns and will alleviate them as follows:
> > > > >
> > > > > - Since AC, SCAn and SPECTRE fail against TaCT on GTSRB, the reviewer is concerned that whether latent space separation holds for this case. **Our answer is positive** --- even Spectral Signature (SS), the earliest and most classical latent separation-based defense, can eliminate over 97.5% of TaCT's poison samples on GTSRB. This is a piece of strong evidence that TaCT still suffers from latent separation on GTSRB.
> > > > > - **AC, SCAn, and SPECTRE are less effective in detecting TaCT on GTSRB is not because this attack does not suffer from the latent space separation**. As we mentioned in our previous rebuttal, many of the existing defenses optimize their hyper-parameters for some specific datasets (e.g., CIFAR-10). During our experiments, we found that the performance of AC, SCAn, and SPECTRE are not very stable. We have tried but failed to find the optimal hyper-parameters for these defenses on GTSRB. This is probably the main reason why these three defenses fail to defend against TaCT on GTSRB.
> > > > > - Finally, in terms of resistence against latent-separation-based defenses, comparing with other attacks, our Adaptive Attacks have the lowest elimination rate under the worst cases (i.e. against the strongest defense out of the four): Adap-Blend (43), Adap-Path (74.4), Blend (84.8), Dynamic (96.2), TaCT (97.5), BadNet (100).

---

> > ### Comment · Reviewer_XjVu · 2022-11-20
> > **Thank you for the response on similarity to Peng et al**
> >
> > Thank you for your response and acknowledgment of the similarity between the proposed method and Peng et al's method.
> >
> > * Thank you for clarifying the differences, which is essential to demonstrate the technical novelty of the paper (although Peng et al's work is an arXiv submission).
> >
> > * As discussed by the authors, one major difference between this work and Peng et al's is the latent space study. This is also the novelty of the paper. Thus, I believe that this difference should be discussed in the paper, and the differences between the two methods should also be discussed in detail to demonstrate the technical contributions of the paper. One of my major concerns previously was that the paper claims the novelty in the methodology while the technique was previously discovered in another work (in spite of an arXiv submission).

---

> > > ### Author Response · Authors · 2022-11-20
> > > **Author Response (Round II)**
> > >
> > > Thank you for your response and clarifications. In particular, thank you for acknowledging our novelty and unique contributions.
> > >
> > > ---
> > >
> > > **Q1**: Thank you for clarifying the differences, which is essential to demonstrate the technical novelty of the paper (although Peng et al's work is an arXiv submission).
> > >
> > > **R1**: Thank you for your acknowledgment of our previous rebuttal. It is a huge encouragement and support to us!
> > >
> > > ---
> > >
> > > **Q2**: As discussed by the authors, one major difference between this work and Peng et al's is the latent space study. This is also the novelty of the paper. Thus, I believe that this difference should be discussed in the paper, and the differences between the two methods should also be discussed in detail to demonstrate the technical contributions of the paper. One of my major concerns previously was that the paper claims the novelty in the methodology while the technique was previously discovered in another work (in spite of an arXiv submission).
> > >
> > > **R2**: Thank you for this constructive suggestion! We have added a detailed comparison between our work and  (Peng et al., 2022) in Appendix F of our revision. We are willing to provide more details if you still have other concerns regarding the differences.
> > >
> > > ---

---

> ### Author Response · Authors · 2022-11-30
> **Thanks to Reviewer XjVu**
>
> We would like to thank you again for reviewing our work and the valuable feedback, and in particular for recognizing the strengths of our paper in terms of motivation, and valuable contributions.
>
> We fully understand the concerns that you raised. During the rebuttal period, we made the following efforts to address them:
>
> 1. We added a thorough discussion on the similarity of our work to the current work by Peng et al. We also highlighted the differences between these two works from many fundamental aspects. In the appendix of our revision, we also added a concurrent statement to clarify the similarity and differences. We hope these efforts can address your concern.
>
> 2. We supplemented additional comparison between our attacks and other attacks on GTSRB. We highlighted that: 1) other attacks indeed suffer from latent separation characteristics even on GTSRB, though some of the latent separation-based defenses are less effective on GTSRB (because their hyperparameters are optimized for CIFAR10). 2) in terms of resistance against latent-separation-based defenses, compared with other attacks, on GTSRB, our Adaptive Attacks still have the lowest elimination rate under the worst cases (i.e. against the strongest defense out of the four): Adap-Blend (43), Adap-Path (74.4), Blend (84.8), Dynamic (96.2), TaCT (97.5), BadNet (100). These results further validate the pervasiveness of latent separation and the advantage of our attacks in terms of mitigating latent separation.
>
> 3. We added additional evaluation results against other defenses that are not based on latent separation, which makes our work more comprehensive.
>
> We hope these efforts can further address your concerns. Please kindly let us know if you have any additional questions or require further clarification. We are happy to address them before the rebuttal ends.

---

> ### Author Response · Authors · 2022-12-05
> **A Gentle Reminder of the Final Feedback**
>
> We would like to thank the reviewer for the helpful discussion during the first round of the review and further comments. We hope our response has adequately addressed your previous comments. We take this as a great opportunity to improve our work and shall be grateful for any additional feedback you could give to us.

---

### Decision · Program_Chairs · 2023-01-20

**Decision:**

Accept: poster

**Justification For Why Not Higher Score:**

The score of the work is perhaps limited.

**Justification For Why Not Lower Score:**

There is clear intellectual contribution.

**Metareview: Summary, Strengths And Weaknesses:**

Backdoor poisoning attacks, in which an adversary manipulates a deep learning model by modifying a small number of training data points, often result in the model learning separable latent representations for poison and clean samples. This latent separation has been widely observed and used as an assumption by defenses designed to identify poison samples. However, the authors of this paper present adaptive backdoor poisoning attacks that challenge this assumption, demonstrating that latent separation is not always a reliable indicator of poison samples. These adaptive attacks maintain a high success rate while minimizing the impact on clean accuracy.

Some reviewers raised concerns about its novelty in the presence of an arXiv paper that has not yet been published. However, the authors provided convincing responses to these concerns and appropriately addressed the comments. The reviewers also had concerns about the limited scope and experimental aspects of the work, but the authors gave convincing responses to these concerns as well.

Based on my own reading of the paper, the openreview comments, and the zoom discussion, it is recommended that this paper be accepted for publication. The authors have addressed the concerns raised by the reviewers, including those related to the novelty of the work and the limited scope and experimental aspects of the study. They have provided convincing responses to these concerns and have made appropriate revisions to the paper. It is recommended that the final version of the paper be accepted for publication after incorporating the comments and suggestions made by the reviewers.


**Note From Pc:**

if the above contains the word "oral" or "spotlight" please see: "oral" presentation means -> notable-top-5% and "spotlight" means -> notable-top-25%. As stated in our emails, we are disassociating presentation type from AC recommendations